# Re-calibrating Feature Attributions for Model Interpretation

**Peiyu Yang**[1]**, Naveed Akhtar**[1]**, Zeyi Wen**[2,3]**, Mubarak Shah**[4]**, and Ajmal Mian**[1]
[1]The University of Western Australia
[2]Hong Kong University of Science and Technology (Guangzhou)
[3]Hong Kong University of Science and Technology [4]University of Central Florida
`{peiyu.yang@research., naveed.akhtar@, ajmal.mian@}uwa.edu.au`
`wenzeyi@ust.hk   shah@crcv.ucf.edu`

## Abstract

The ability to interpret machine learning models is critical for high-stakes applications. Due to its desirable theoretical properties, path integration is a widely used scheme for feature attribution to interpret model predictions. However, the methods implementing this scheme currently rely on absolute attribution scores to eventually provide sensible interpretations. This not only contradicts the premise that the features with larger attribution scores are more relevant to the model prediction, but also conflicts with the theoretical settings for which the desirable properties of the attributions are proven. We address this by devising a method to first compute an appropriate reference for the path integration scheme. This reference further helps in identifying valid interpolation points on a desired integration path. The reference is computed in a gradient ascending direction on the model's loss surface, while the interpolations are performed by analyzing the model gradients and variations between the reference and the input. The eventual integration is effectively performed along a non-linear path. Our scheme can be incorporated into the existing integral-based attribution methods. We also devise an effective sampling and integration procedure that enables employing our scheme with multi-reference path integration efficiently. We achieve a marked performance boost for a range of integral-based attribution methods on both local and global evaluation metrics by enhancing them with our scheme. Our extensive results also show improved sensitivity, sanity preservation and model robustness with the proposed re-calibration of the attribution techniques with our method.[1]

## 1 Introduction

How to interpret deep learning predictions is a major concern for the real-world applications, especially in the high-stakes domains. Feature attribution methods explain a model's prediction by assigning importance scores (attributions) to the input features (Simonyan et al., 2014; Springenberg et al., 2015; Shrikumar et al., 2017). They assert that *features with larger attribution scores are more relevant to the model prediction than those with smaller scores.* Among a variety of attribution methods, integral-based techniques (Sundararajan et al., 2017; Sturmfels et al., 2020) are particularly attractive because they satisfy certain desirable axiomatic properties, which others do not. This also makes them suitable for model regularization (Chen et al., 2019; Erion et al., 2021).

Inspired by the cooperative game theory, integral-based attribution methods introduce a *reference* to represent the *absence* of the input signal. This allows them to calculate the attribution for the *presence* of an input feature (Sundararajan et al., 2017; Erion et al., 2021; Pan et al., 2021). Since the attribution is computed with respect to a reference, finding the correct reference that represents the absence of a feature in a true sense, is critical for the reliability of these methods. In their Integrated Gradient (IG) method, Sundararajan et al. (2017) chose a black image (zero input) as the reference. However, this result is always assigning zero attribution to black input pixels. Sturmfels et al. (2020) confirmed that a fixed reference renders an attribution method blind to the features that the reference

---

[1]Our code is available at `https://github.com/ypeiyu/attribution_recalibration`

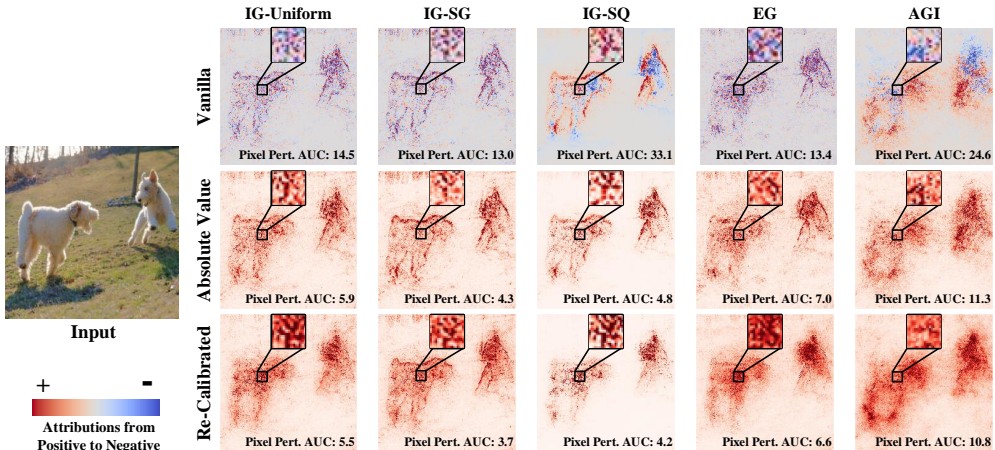

Figure 1: Attribution maps of existing methods (Vanilla) suffer from large variations between positive and negative scores, which forces them to use the Absolute Values of the scores for a sensible interpretation. However, this contradicts the primary assertion of this research direction, and with the settings used to claim the desirable theoretical properties of these methods. Re-calibrating the methods with our technique addresses the issue, while also improving the performance. Maps are shown for VGG-16. The pixel perturbation (%) AUC is reported. Lower values are more desirable.

employs. To resolve that, multiple references are now commonly used by the attribution methods. The choices of multiple references include, uniform noise references (IG-Uniform) (Sturmfels et al., 2020), Gaussian noise references (IG-SG and IG-SQ) (Smilkov et al., 2017; Hooker et al., 2019), training distribution (EG) (Erion et al., 2021) and adversarial examples (AGI) (Pan et al., 2021), etc. However, the attribution maps computed by these methods face frequent variations between positive and negative attribution scores, see Figure 1 top row. To handle that, these methods must use absolute scores to generate plausible explanations, see Figure 1 middle row. Numerically, computing the absolute values re-orders the estimated attributions for the pixels, which violates the primary assertion of the attribution methods. Moreover, relying only on the magnitude of the attributions is also in contradiction with the axiomatic properties, which are proven for the actual numerical scores. This inconsistency compromises the desirability of these methods.

To address these problems, we develop a method to compute the desired reference for an input along the model's gradient ascending direction. Moreover, we allow the gradient integration along a nonlinear path from the reference to the input. This is made possible by systematically identifying valid interpolation points on the path. It eventually enables us to directly use the actual, instead of the absolute attributions for model interpretation. We further devise a technique to efficiently compute the integral with valid references which can be estimated using the predefined references employed by the existing methods. This enables us to leverage our technique to re-calibrate the attributions of the existing methods without additional computational overhead. Figure 1 bottom row shows the attribution maps calculated by the popular integral-based attribution methods calibrated with our technique. These maps are computed with the actual attributions scores, not the absolute values. Hence, they conform to the primary assertion of the attribution methods and to the settings used in establishing their theoretical properties. Moreover, they achieve better quantitative scores.

In our experiments, quantitative evaluation is performed with pixel perturbation (Samek et al., 2016) and DiffROAR (Shah et al., 2021) on ImageNet-2012 validation set (Russakovsky et al., 2015), CIFAR-100 and CIFAR-10 (Krizhevsky et al., 2009). We show a marked performance improvement for a range of integral-based attribution methods by re-calibrating them with our technique. We also provide a detailed sensitivity analysis of the improved methods with Sensitivity-n (Ancona et al., 2018), and passing the sanity checks (Adebayo et al., 2018). Moreover, we also show consistent improvements in attribution prior based regularization (Erion et al., 2021) with our technique. A considerable performance gain for a variety of techniques and models across a range of evaluation metrics ascertains the effectiveness of our re-calibration method. In summary, we make the following major contributions.

1. We propose a method to compute a reliable reference for integral-based attribution methods.

2. We devise a technique to efficiently integrate over a non-linear path that enables meaningful interpretations using the actual, as opposed to absolute attribution scores.

3. We enhance a range of existing methods with our findings to demonstrate a considerable performance boost across the board over multiple evaluation metrics, and for model regularization.

## 2 RELATED WORK

Feature attribution methods explain a model's prediction by assigning attribution scores to the input features. These techniques can be roughly categorized into perturbation based, and back-propagation based methods. The former, e.g., (Zeiler & Fergus, 2014; Fong & Vedaldi, 2017), calculates the attribution scores by repeatedly perturbing the input features and analyzing the resulting effects on the model output. Back-propagation methods, e.g., (Simonyan et al., 2014; Sundararajan et al., 2017) estimate the attribution scores using a backward pass of model gradients, and enjoy multiple advantages over the former (Ross et al., 2017; Chen et al., 2019). Since our contribution relates to the back-propagation methods, we focus on the related work in this sub-category.

**Gradient-Based Methods:** Simonyan et al. (2014) first employed input gradients for class-specific saliency mapping for model interpretation. Deconvolutional Network (Zeiler & Fergus, 2014) and Guided Backpropagation (Springenberg et al., 2015) also propagate the activation information in the backward direction to highlight important input features. Layer-wise Relevance Propagation (Bach et al., 2015) defines the relevance of neurons to recursively assign the attributions to the input. Similarly, DeepLIFT (Shrikumar et al., 2017) assigns attribution scores to the input by comparing the relative contributions of features using a reference input.

**Integral-Based Methods:** Integrated Gradients (IG) (Sundararajan et al., 2017) computes an integral for the input attribution along a linear path from the input to a reference representing the absence of the input feature. However, the zero reference chosen by the IG makes it blind to black pixels. To overcome that, Expected Gradients (Erion et al., 2021) samples multiple references from the model's training distribution. IG-SG (Smilkov et al., 2017) and IG-SQ (Hooker et al., 2019) perturb input images with Gaussian noise as the references. In addition, Adversarial Gradient Integration (Pan et al., 2021) adversarially perturbs the input to compute the desired reference. Sturmfels et al. (2020) compared different references for the path attribution methods, demonstrating their weaknesses.

**Evaluation Metrics:** A fair benchmarking is critical to establish the reliability of attribution methods. Pixel perturbation (Samek et al., 2016; Petsiuk et al., 2018; Srinivas & Fleuret, 2019) is one of the popular quantitative evaluation techniques, which removes the most or least salient pixels (as deemed by the attribution method) to note their effects on model output changes. It is operated on individual inputs, hence considered a local metric. ROAR (RemOve And Retrain) (Hooker et al., 2019) and DiffROAR (Shah et al., 2021) are popular global metrics that utilize retraining over data distribution after removing the attributed pixels from the data. Sensitivity-n (Ancona et al., 2018) is proposed to test the sensitivity of different feature groups while accounting for the attribution scores. Adebayo et al. (2018) also proposed sanity checks to verify reliability of the attribution maps computed by different methods. All these metrics evaluate different aspects of the attribution techniques. We employ all of them for an extensive benchmarking.

## 3 PRELIMINARIES

For classification, a machine learning model maps an input $x = [x_1, \ldots, x_n] \in \mathbb{R}^n$ to an output score $S_c(x)$ for class $c$. For explaining the model prediction, an attribution method $M$ aims at attributing this score to the input features (pixels) by an attribution map $M^c(x) = [M_1^c(x), \ldots, M_n^c(x)]$.

**Input Gradients:** To explain model predictions, Input Gradients (Simonyan et al., 2014) computes model gradients with respect to the input $x$ as the importance scores (attributions). Following the notations from above, for class $c$, the attribution of the $i$-th feature $x_i$ calculated by this method is

$$M_i^c(x) = \partial S_c(x)/\partial x_i. \qquad (1)$$

Simonyan et al. (2014) arrived at this strategy by applying the first-order Taylor expansion to the non-linear model: $S_c(x) \approx w^T * x + b$. This allows the attribution of each input feature to be

represented by the input gradient $w_i = \partial S_c(x)/\partial x_i$. However, due to the high non-linearity of deep models, the input gradients cannot provide reliable attributions. Ancona et al. (2018) also regarded the input gradients as a local attribution method that cannot provide a complete explanation for a specific prediction. Input gradients can mainly indicate *how to obtain the desired output by changing the features around the original input*. This notion also has similarities to adversarial examples and other feature visualization methods (Goodfellow et al., 2015; Carter et al., 2019).

**Integrated Gradients:** Different from the input gradients with local interpretability, integral-based attribution methods, e.g., Integrated Gradients (IG) (Sundararajan et al., 2017), estimate an integral over a path to calculate the attributions. Inspired by the cooperative game theory, IG chooses a reference to represent the complete absence of the input signals, which enables it to integrate gradients from the reference to calculate the attribution for the presence of the input features. The IG calculates the attribution for the $i$-th input feature $x_i$ with the reference $x' = [x'_1, \ldots, x'_n] \in \mathbb{R}^n$ as follows

$$M_i^c(x, x') = (x_i - x'_i) \cdot \int_{\alpha=0}^1 \frac{\partial S_c(\tilde{x})}{\partial \tilde{x}_i}\bigg|_{\tilde{x}=x'+\alpha(x-x')} \mathrm{d}\alpha, \tag{2}$$

where $\alpha$ represents the step along a linear path from the reference to the input. Since the integral of Equation (2) satisfies the fundamental theorem of calculus, IG can satisfy the desirable completeness axiom, i.e., the output changes can be completely attributed to the input features: $\sum_{i=1}^n M_i^c(x, x') = S_c(x) - S_c(x')$. In contrast to the input gradients, integral-based attribution methods are regarded as global attribution methods (Ancona et al., 2018), which can *estimate the marginal effect from the reference to the input for model prediction*.

**Existing Inconsistency:** The primary assertion of the attribution methods is that *the input features with larger attribution scores are more relevant to the model predictions*. However, the attributions computed by the back-propagation techniques suffer from frequent variations between positive and negative scores, even among the neighboring pixels of a (relatively) uniform image region. Thus, for plausible interpretations, these methods eventually use the absolute values of the attributions instead of the actual numerical scores (Sundararajan et al., 2017; Srinivas & Fleuret, 2019). However, considering the absolute values not only re-orders the computed pixel importances, it also contradicts the theoretical settings assumed to prove the desirable axiomatic properties of these techniques.

## 4 METHODOLOGY

In this section, we first assume a reliable attribution calculation with an automatically chosen reference. Then we propose a technique to estimate the proposed attribution, which enables us to reveal a reason for the inconsistency problem. Finally, the proposed technique is extended to popular attribution methods for calibrating their attributions within limited computational resources.

### 4.1 PURSUING RELIABLE ATTRIBUTION

The reference image in an integral-based method has the central importance. As noted earlier, the zero image of IG (Sundararajan et al., 2017) renders it blind to the black pixels in the input. The primary motivation behind using multiple references created with uniform noise (Sturmfels et al., 2020), Gaussian noise (Smilkov et al., 2017) or training samples (Erion et al., 2021), etc., is to avoid this blindness issue. These methods create multiple references in the hope that some of those can correctly capture the abstract notion of feature absence, thereby resulting in a reliable attribution.

Instead of hoping to accidentally stumble on the desired reference, we propose to systematically identify it. To that end, we can modify the input image with the model gradients to construct the reference. Optimizing the input in the gradient ascending direction leads to local features that decrease the target prediction scores, providing a reasonable analogy of feature 'absence' in computational sense. Such a reference is computed with respect to the model itself, instead of relying on an external operator, which can also ensure the model-fidelity of the eventually computed attribution scores. We also emphasize that considering a uniform reference for all the input features is not ideal, as feature absence is a relative notion which is not handled well by uniformity assumption. Hence, a set of references for an input is more desirable, which we also employ.

Given a feature $x_i$ of the input $x$, we hypothesize a computable desired reference $x' = D_i$ within a reference set $D$. Then, the attribution $M_i^c$ of $x_i$ can be calculated by integrating the gradients along

the integration path from the input $x$ to the reference $x'$ as

$$M_i^c(x, x') = (x_i - x_i') \times \int_{\alpha=0}^{1} \left. \frac{\partial S_c(\tilde{x})}{\partial \tilde{x}_i} \right|_{\tilde{x} = x' + \gamma_i(\alpha)(x - x')} d\alpha \quad \text{s.t.} \quad x' = D_i, \tag{3}$$

where $\gamma_i(\cdot)$ indicates a (possibly non-linear) integration path function defined by $x$ and $x'$, with the step $\gamma_i(\alpha)$ over a domain $\Omega \in \mathbb{R}^n$, and $\tilde{x}$ is the interpolation point that resides on this path. We note that our idea of computable reference along the gradient ascending direction has some similarity with the Adversarial Gradient Integration (Pan et al., 2021) which integrates gradients along the path from the input to class-specific adversarial examples. However, in contrast to AGI, we allow different features to have different paths to different references. This effectively enables integration along a non-linear path defined by the reference set.

## 4.2 Integral Estimation with Desired Interpolations

So far, we have only set our compass in the direction of pursuing reliable attributions. Different from the linear path employed by the IG-like methods (Sundararajan et al., 2017; Erion et al., 2021), this pursuit requires integration of gradients along a non-linear path $\gamma_i$, as per Equation (3).

To compute the required integral, the key is to identify the interpolation points that reside on the integration path from the desired reference $x'$ to the input $x$ (or vice versa). Assuming an interpolation point $\tilde{x}$ to be in the same input space $\mathbb{R}^n$, it is possible to identify it systematically. Recall that, the gradient can induce a change in the input feature towards the desired reference, as discussed in the previous section. Given an input feature $x_i$ of input $x$, the signed input gradient within the set $\{\nabla_{x_i}^+, \nabla_{x_i}^0, \nabla_{x_i}^-\}$ indicates the direction of the variation $\delta_i$ from $x_i$ to its local desired reference $x' = D_i$ along the gradient ascending direction. Thus, the gradient with respect to an interpolation point $\tilde{x}$ residing on the path between the input and the reference, has the same sign as the variation $\delta_i \in \{\delta_i^+, \delta_i^0, \delta_i^-\}$ over the $i$-th feature of $\tilde{x}$. If the interpolation point does not reside on the desired path, the gradient $\nabla_{\tilde{x}_i}$ always has the opposite sign compared to $\delta_i$. To exemplify, let an input feature $x_i$ have a positive gradient $\nabla_{x_i}^+$, and this indicates $x_i$ should take a negative step towards its local desired reference ($x_i - \delta_i^+ = x_i'$). This implies that the variation $\delta_i$ is positive, i.e., $\delta_i^+$. As such, we can identify the interpolation point $\tilde{x}$ that resides on our desired integration path by the product between the gradient $\nabla_{\tilde{x}_i}$ and the variation $\delta_i$.

In light of the above discussion, we can integrate gradients over identified interpolations. If we can obtain the variation between $x_i$ and $x_i'$, the required integral can be estimated. Defining the interpolations as a set $\tilde{X} = \{\tilde{x}^{(1)}, \dots, \tilde{x}^{(m)}\}$ with $m$ interpolations residing on the integration path, we can obtain the variation from $x_i$ to $x_i'$ by following Lemma 1. We use $\tilde{x}^{(j)}$ instead of $\tilde{x}$ here to distinguish multiple interpolated image points, and $\tilde{x}_i^{(j)}$ is the $i$-th feature of $\tilde{x}^{(j)}$.

**Lemma 1.** *Given two feature points $x_i$ and $x_i'$, and $m$ uniformly distributed interpolation points $\tilde{x}_i^{(1)}, \dots, \tilde{x}_i^{(m)}$ between $x_i$ and $x_i'$, the variation $\delta_i = x_i - x_i'$ is proportional to the average distance from $x_i$ to all interpolation points $\tilde{x}_i$, and the proportionality coefficient is 2.*

We provide proof of Lemma 1 in Appendix A.1. Following Lemma 1, the attribution $M_i$ defined in Equation (3) can be estimated over $m$ interpolations as follows

$$M_i(x, \tilde{X}) = \frac{\lambda}{m} \sum_{j}^{m} \delta_i^{(j)} \nabla_{\tilde{x}_i^{(j)}}, \tag{4}$$

where $\lambda$ is the proportionality coefficient, and $\delta_i^{(j)}$ is the variation between $x_i$ and $\tilde{x}_i^{(j)}$.

The above insights are simple yet powerful. They can be leveraged to re-calibrate attribution scores computed by the existing path attribution methods. Current integral-based attribution methods with multiple references, e.g., IG-SG, EG and AGI, employ different strategies to sample the references and integrate the gradients of interpolation points from the input to the references. These points can be viewed as adopting different sampling strategies to approach the input space via Monte Carlo estimation (Erion et al., 2021). This view allows us to utilize our technique with these attribution methods to modulate their interpolations for the integral estimation. In the text to follow, we distinguish the interpolation points resulting from incorporating our insights into an attribution technique as *valid* interpolations. In essence, current methods end up estimating attributions using interpolation points that are not valid, which causes them to compute sub-optimal attributions.

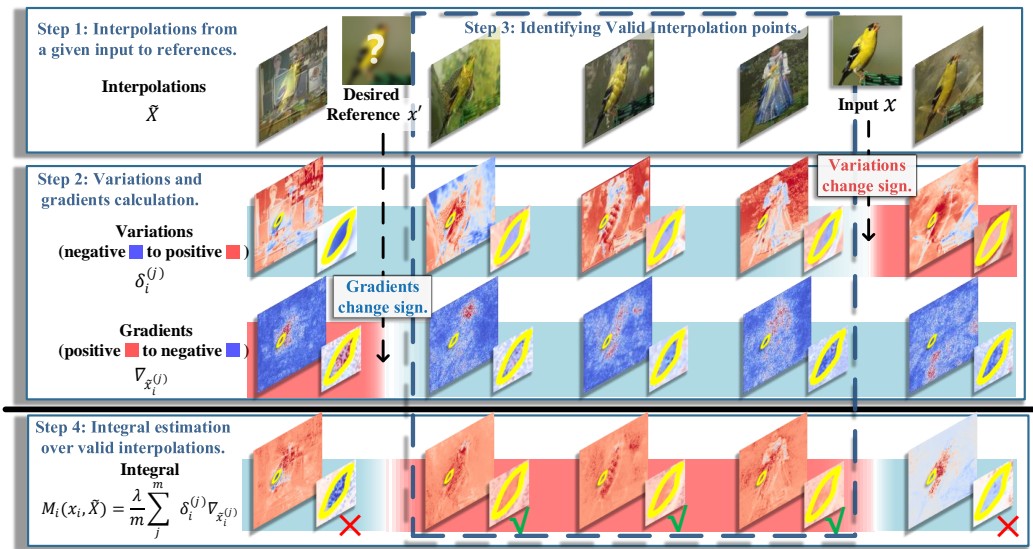

Figure 2: Illustration of integration over valid interpolations. *Step 1*: For an input $x$, multiple interpolated images are computed with given references. *Step 2:* Variations and gradients are computed for the interpolated images. *Step 3:* Variations become positive when they cross-over the input, and gradients become positive when they over-shoot the desired reference. Between these two extremes, interpolation points remain valid. Hence, they are selected. *Step 4:* Only valid interpolation points are used for integration, and the rest are discarded.

In Figure 2, the notion of integration using valid interpolation points is illustrated. The central idea of identifying the valid interpolations for any reference, followed by the integration with Equation 4 is generic. In the figure, for the input $x$, $x'$ is the desired reference in a gradient ascending direction. The first step produces interpolation points to different references. Then, the variations $\delta_i^{(j)}$ and the corresponding gradients $\nabla_{\tilde{x}_i^{(j)}}$ are calculated in the second step. The red and blue colors indicate the positive and negative values in $\delta_i^{(j)}$ and $\nabla_{\tilde{x}_i^{(j)}}$ whose sign change after crossing the input feature $x_i$ and the desired reference $x'_i$ separately. The variation and the gradient always have the same signs for the valid interpolation points along the integration path in the neighborhood of the input. This observation is used to identify the valid interpolations. Finally, the attribution $M_i$ is computed over the identified valid interpolations following Equation 4.

---

**Algorithm 1:** Attribution Re-Calibration

**input** : Input $x$, reference set $D$,
  number of steps $n$.
**output:** Attribution $M_i$

1 **initialize**: $\mathcal{M} \leftarrow \emptyset$
2 **for** *each $x'$ in $D$* **do**   // Iterate references.
3     **for** $k \leftarrow 1$ **to** $n$ **do**   // Integral path.
4        $\tilde{x} \leftarrow x' + \frac{k}{n}(x - x')$
5        $\nabla_{x_i} \leftarrow \frac{\partial S_c(\tilde{x})}{\partial \tilde{x}_i}$
6        **if** $\nabla_{x_i} \cdot (x_i - \tilde{x}_i) \geq 0$ **then**
7           // Integrate gradients with
          valid interpolations.
8           $\mathcal{M} \leftarrow \mathcal{M} \cup (x_i - \tilde{x}_i) \times \nabla_{x_i}$
9 $M_i \leftarrow \lambda \times \text{avg}(\mathcal{M})$

---

We also present the pseudo-code to re-calibrate the current integral-based methods with valid interpolations in Algorithm 1. Given a feature $x_i$ of the input $x$, and a reference set $D$ identified by the method's own strategy, we iterate over the references (Line 2). For each reference, the method computes the gradients of interpolated images $\tilde{x}$ from the input to the reference (Lines 3-5). However, different from the existing practice, we let the integral to be estimated on valid interpolations that satisfy the condition that $\nabla_{x_i} \cdot (x_i - \tilde{x}_i)$ is positive (Line 6-8). Finally, we compute the integral with the averaging operation (Lines 9).

### 4.3 Efficient Attribution Re-calibration with Valid References

Sampling sufficient references and interpolation points allow us to estimate an accurate integral. However, limited computational resources must be accounted for by the attribution methods. To

estimate an accurate integral with limited resources, we employ importance sampling (Kloek & Van Dijk, 1978). Let $f(\cdot)$ be a function of $n$ inputs $X = \{x^{(1)}, \ldots, x^{(n)}\}$ over the domain $\Omega \in \mathbb{R}^n$. Then, the integral is defined as $\boldsymbol{v} = \int_\Omega f(X)dX$. Importance sampling approaches this integral as

$$\boldsymbol{v} = \int_\Omega \frac{f(X)}{g(X)} g(X) dX, \tag{5}$$

where $g(X) > 0$ for $X \in \Omega$ is a probability function called the importance function. In order to estimate the integral, we generate $n$ samples $x^{(i)}$ from $g(X)$ and estimate the integral by the following sample-mean formula.

$$\hat{\boldsymbol{v}} = \frac{1}{n} \sum_{i=1}^{n} \frac{f(x^{(i)})}{g(x^{(i)})}, \tag{6}$$

where the estimator $\hat{\boldsymbol{v}}$ can converge to the integral $\boldsymbol{v}$. Instead of randomly sampling inputs in the input space, importance sampling enables us to sample input with the importance function for efficiently estimating the integral. Let $f(X)/g(X)$ be a function $\beta(X)$, we have the following Lemma.

**Lemma 2.** *If the ratio between the function $f(X)$ and $\beta(X)$ has small fluctuations, the estimator $\hat{\boldsymbol{v}}$ can be approached by the mean formula $\hat{\boldsymbol{v}} = \frac{1}{n} \sum_{i=1}^{n} \beta(x^{(i)})$, where $x^{(i)}, \ldots, x^{(n)}$ are sampled uniformly from $g(x)$.*

Let $\beta(X)$ be defined on a linear integration path from the input to the reference. If the underlying distribution of $g(X)$ fluctuates slightly, we can estimate the integral along a linear path with $\beta(X)$. Since only slight fluctuations cannot be guaranteed along the full path in our case, we approach $f(X)$ with $\beta(X) \cdot g(X)$ in different segments. Since the integral estimated from an input to its desired reference is also expected to have a positive attribution, we relax the condition of the integral estimated on the valid interpolations with $\nabla \cdot \delta \leq 0$ to the valid reference that has a positive integral estimation for approaching a segment in $f(X)$. Then, we average positive integrals in $k$ segments for estimating the final integral as

$$M_i = \frac{\lambda}{k} \sum_{i}^{k} \int_{\alpha=0}^{\frac{k}{m}} \left. \frac{\partial S(\tilde{x})}{\partial \tilde{x}_i} \right|_{\tilde{x}=x'+\alpha(x-x')}, \tag{7}$$

The integral estimation is also guided by Lemma 1. Using the same number of references, the proposed technique enables attribution methods to calculate more precise attributions without incurring needless computational overhead. In Appendix A.2, we provide the pseudo-code of our technique.

## 5 EXPERIMENTS

To evaluate the proposed technique, we conduct experiments to test both performance and sensitivity. We take existing popular multi-reference attribution techniques as baseline methods; including IG-SG (Smilkov et al., 2017), IG-SQ(Hooker et al., 2019), IG-Uniform (Sturmfels et al., 2020), EG (Pan et al., 2021) and AGI(Pan et al., 2021). All these methods enhanced with the proposed re-calibration technique are benchmarked in our experiments. Additionally, other feature attribution methods, including InputGrad (Simonyan et al., 2014), Integrated Gradients (IG) (Sundararajan et al., 2017), SmoothGrad (SG) (Smilkov et al., 2017) and FullGrad (Srinivas & Fleuret, 2019) are also compared in our evaluation. More detailed hyper-parameter settings and extended experimental results are provided in Appendices A.3 and A.4.

### 5.1 PIXEL PERTURBATION PERFORMANCE

Pixel perturbation is a widely used quantitative evaluation metric for the attribution methods (Samek et al., 2016; Ancona et al., 2018; Srinivas & Fleuret, 2019; Yang et al., 2023). Since an attribution map is expected to identify the correct order of relative importance of the input features, pixel perturbation evaluation iteratively removes the most or the least salient pixels (Pixel Insertion and Deletion) and measures the output changes to quantify the performance of attribution methods. In this part, we employ DiffID metric (Yang et al., 2023) to measure the performance difference on images with inserted $k\%$ most salient pixels and deleted $1 - k\%$ least salient pixels.

Figure 3 reports the fractional output change difference with pixel deletion and insertion on ImageNet 2012 Validation set (Russakovsky et al., 2015) evaluated on ResNet-34 (He et al., 2016b) and

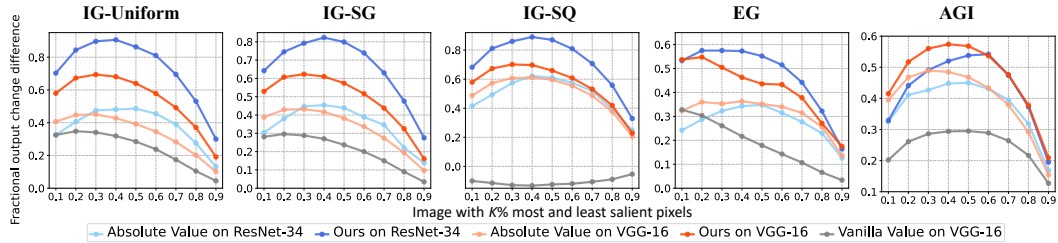

Figure 3: Results on DiffID metric on ImageNet 2012 val. set. Higher values indicate better results. Enhancing methods with our technique provides a considerable performance boost.

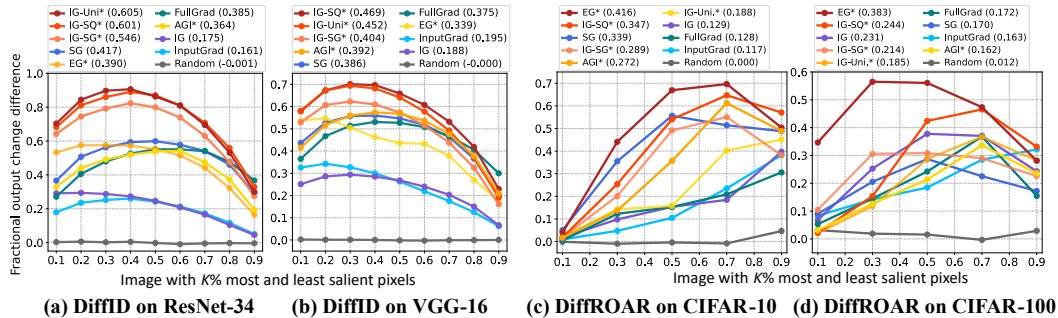

Figure 4: DiffID results comparison of enhanced baselines with other existing methods using ResNet-34 **(a)**, and VGG-16 **(b)** on ImageNet 2012 val. set. Baseline methods enhanced with our technique are identified with **\***. AUC values for each curve is reported. Larger values are more desirable. DiffROAR results on CIFAR-10 **(c)** and CIFAR-100 **(d)** are also reported following the same convention. Larger AUC values imply better results.

VGG-16 (Simonyan & Zisserman, 2015). Higher values in these curves indicate better performance. The Absolute Value curves are obtained by allowing the methods to replace the negative attributions with their absolute values. For the Vanilla values, we do not allow this operation. The results are reported for five popular baseline multi-reference integral-based attribution methods. We can observe that the proposed re-calibration technique consistently improves all the baseline methods by a large margin. It is also clear that the original methods heavily rely on taking the absolute value to provide explanations. The proposed technique strongly improves IG-Uniform, IG-SG and AGI. The improvements for EG and IG-SQ are also reasonably large. These results ascertain that our technique can enable an across-the-board performance gain for the existing methods.

In Figure 4a and 4b, we provide further comparisons on DiffID baseline with the popular original exiting methods, including InputGrad, IG, SG and FullGrad. A noticeable performance gap is clearly visible between these methods and the enhanced baselines, favoring the enhanced techniques. The results show that our method provides a notable positive off-set to the state-of-the-art in integral-based attribution.

## 5.2 REMOVE AND RETRAIN PERFORMANCE

Although pixel perturbation can provide an estimate of the interpretability of the attribution methods, Hooker et al. (2019) argued that pixel perturbations also cause a distribution shift for the inputs, and thus it is unclear whether the output change comes from removing informative pixels or the distribution shift. To alleviate the effect of input distribution shift, RemOve and Retrain (ROAR) (Hooker et al., 2019) is a commonly used metric that retrains the model with the perturbed images produced by the pixel perturbation and measures the output change. To provide a more comprehensive evaluation, DiffROAR (Shah et al., 2021) extends the ROAR metric to test the difference between the model retrained with the $k\%$ most and least salient pixels. To thoroughly establish the effectiveness of our technique, we also conduct experiments using DiffROAR. We again enhance the popular attribution methods IG-Uniform, IG-SG, IG-SQ, EG and AGI for this evaluation.

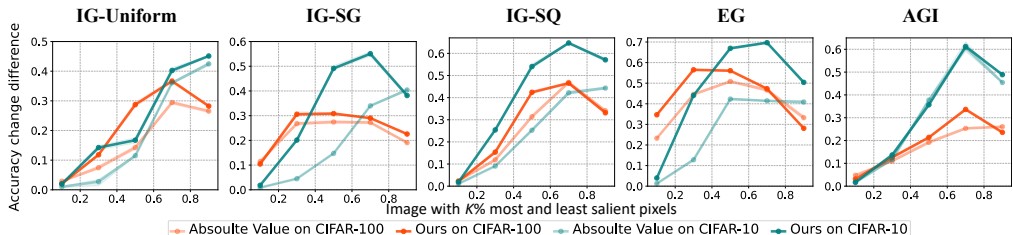

Figure 5: Results on DiffROAR metric on ImageNet 2012 val. set. Larger values indicate better performance. Methods achieve a consistent gain with our enhancement.

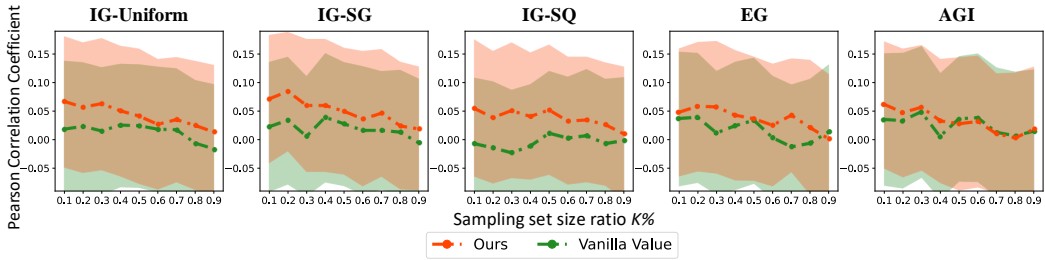

Figure 6: Sensitivity-n evaluation. Higher values indicate better performance.

We retrain the PreActResNet-18 (He et al., 2016a) on both CIFAR-100 and CIFAR-10 datasets (Krizhevsky et al., 2009). Figure 5 shows the accuracy change difference on the model retrained with the most and least important $k\%$ pixels separately. Since retraining models with different perturbed inputs causes these models to fit different input distributions, we can observe that the curves in Figure 5 have different undulations as compared to the pixel perturbation experiments. Nevertheless, the results establish that our method again generally improves the performance for all the methods. We also summarize our quantitative evaluation using DiffROAR in Figure 4c and 4d for CIFAR-10 and CIFAR-100 datasets. The figure compares the results with other existing methods which are used in their original form. These results also consistently favor our technique of enhancing the methods.

### 5.3 SENSITIVITY-N PERFORMANCE & FURTHER RESULTS

Ancona et al. (2018) proposed Sensitivity-n metric to verify that the model output variations are sensitive to the features as prescribed by the attribution method. Given any subset of features $x_S = [x_1, \ldots, x_m] \subseteq x$, Sensitivity-n requires the sum of attributions $M_i$ to hold $\sum_{i=1}^m M_i = S_c(x) - S_c(x_{[x_S=0]})$. Although no method can be expected to fully satisfy Sensitivity-n due to practical reasons, the definition still enables a pragmatic benchmarking scheme. By varying the feature fraction in the subsets in the range [0.1, 0.9] of the total features, we test the Pearson Correlation Coefficient (PCC) computed between the sum of the attributions and the target output variation, shown in Figure 6. For each feature subset, we sample 100 different subsets of the input features with a uniform probability distribution. The PCC is averaged over 1000 testing images of ImageNet 2012 Validation set. Compared with the vanilla attribution map, we can observe that the proposed re-calibration generally shows better sensitivity-n results for all the baseline methods. In Appendix A.4.3, we provide more results for the Sensitivity-n metric.

## 6 CONCLUSION

We identified an inconsistency between the theoretical treatment of path integral-based attribution and its practical implementation in various methods. We address this by devising a scheme that re-calibrates attribution computation by identifying a desired reference and selecting only valid interpolation points for integration. This scheme is applicable to all integral-based attribution methods. Enhancing a range of multi-reference methods with it, we showed considerable performance gain on multiple evaluation metrics. We also demonstrated sanity preservation and improved model robustness by using the proposed scheme as attribution prior regularization.

## ETHIC STATEMENT

Explaining the model prediction is highly relevant to the safety and privacy of users. In this paper, we improve the back-propagation based feature attribution methods. The improved attribution methods need to fully access the model and inputs to produce explanations. Therefore, it is critical to carefully protect the safety of the training data and trained models for practical applications.

## REPRODUCIBILITY STATEMENT

The detailed experimental setup including the hyper-parameter choice of compared attribution methods on different datasets with the specific experimental framework and platform is mentioned for reproducing our experiments in Appendix A.3. In addition, the implementation code can be found in the supplementary material.

## ACKNOWLEDGMENT

This research was supported by ARC Discovery Grant 190102443. Professor Ajmal Mian is the recipient of an Australian Research Council Future Fellowship Award (project number FT210100268) funded by the Australian Government. Dr. Naveed Akhtar is the recipient of Office of National Intelligence, National Intelligence Postdoctoral Grant (project number NIPG-2021-001) funded by the Australian Government.

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

# A APPENDIX

## CONTENTS

### A.1 PROOF

In this section, we provide the proof of Lemma 1 and Lemma 2.

*Proof of Lemma 1.* Given two points $x_i$ and $x'_i$, $n$ interpolations $\tilde{X} = \{\frac{1}{n}(x_i - x'_i), \ldots, \frac{n}{n}(x_i - x'_i)\}$ uniformly distributed between $x_i$ and $x'_i$. The average distance $\bar{d}$ from $x_i$ to all interpolation points is calculated as follows.

$$
\begin{aligned}
\bar{d} &= \sum_{a=1}^{n} \frac{a(x_i - x'_i)}{n} \frac{1}{n}, \\
&= (\frac{1}{n} + \cdots + \frac{n}{n}) \frac{(x_i - x'_i)}{n}, \\
&= \frac{1+n}{2n}(x_i - x'_i).
\end{aligned}
\tag{8}
$$

Then, the limit of $\frac{1+n}{2n}(x_i - x'_i)$, as the number $n$ of interpolations approaches positive infinity, is $\frac{1}{2}(x_i - x'_i)$, donated as follows.

$$
\lim_{n \to +\infty} \frac{1+n}{2n}(x_i - x'_i) = \frac{1}{2}(x_i - x'_i)
\tag{9}
$$

Thus, the distance $\delta = x_i - x'_i$ between $x_i$ and $x'_i$ is proportional to the the average distance $\bar{d}$ with coefficient 2 donated as $\delta = 2\bar{d}$. □

*Proof of Lemma 2.* If the ratio $f(x)/\beta(x)$ fluctuates slightly, the importance function $g(x)$ from which we sample the inputs can be approximatively viewed as a uniform probability density function. Therefore, the importance scores $\beta(X_1), \ldots, \beta(X_n)$ can be calculated with the uniformly sampled inputs $X_1, \ldots, X_n$ as follows.

$$
\beta(X_i) = \frac{f(X_i)}{g(X_i)},
\tag{10}
$$

Thus, the $\beta(X_i)$ can be used to estimate the integral $\hat{v}$. □

### A.2 PSEUDO-CODE TO RE-CALIBRATION ATTRIBUTION WITH VALID REFERENCES

In this section, we present the pseudo-code of our technique to re-calibrate the integral-based methods with valid references in Algorithm 2. Given a feature $x_i$ of the input $x$, and a reference set $D$ identified by the method's own strategy, we iterate over the references (Line 2). For each reference, the method computes an average $\bar{g}$ of gradients with respect to the interpolated image $\tilde{x}$ from the input $x$ to the reference $x'$ (Lines 3-6). Different from the existing methods, we let the integral to be estimated on the valid references that satisfy the condition that $\bar{g} \cdot (x_i - x'_i)$ is positive (Lines 7-9). Finally, we approximate the integral with the averaging operation (Line 10). As such, We can re-calibrate the integral-based methods by identifying valid references with valid references without the additional computational overhead.

---

**Algorithm 2:** Attribution Re-Calibration

**input** : Input $x$, reference set $D$, number of steps $n$.
**output:** Attribution $M_i$

1 **initialize**: $\mathcal{M} \leftarrow \emptyset$, $\bar{g} \leftarrow 0$
2 **for** *each $x'$ in $D$* **do** // Iterate references.
3    **for** $k \leftarrow 1$ **to** $n$ **do** // Integral path.
4       $\tilde{x} \leftarrow x' + \frac{k}{n}(x - x')$
5       // Compute the average gradient.
6       $\bar{g} \leftarrow \bar{g} + \frac{\partial S_c(\tilde{x})}{\partial \tilde{x}_i}/n$
7    **if** $\bar{g} \cdot (x_i - x'_i) \geq 0$ **then**
8       // Integrate integrals estimated with valid references.
9       $\mathcal{M} \leftarrow \mathcal{M} \cup (x_i - x'_i) \times \bar{g}$
10 $M_i \leftarrow \lambda \times \text{avg}(\mathcal{M})$

---

### A.3 EXPERIMENTAL SETUP

In this section, we provide details of the experimental setup, including the hyperparameter choice and the experimental software and platform.

### A.3.1 HYPERPARAMETER CHOICE

**Hyperparameter Choice of Attribution Methods.** In our experiments, we employ five popular integral-based attribution methods as our baseline methods including IG-Uniform (Sturmfels et al., 2020), IG-SG (Smilkov et al., 2017), IG-SQ (Hooker et al., 2019), EG (Erion et al., 2021) and AGI (Pan et al., 2021). Specifically, IG-Uniform defines multiple by sampling references from uniform noise whereas IG-SQ squares the gradients in the integral estimation. Similarly, IG-SG adds Gaussian noise to the input as references. EG samples training images as the reference to assume the underlying training distribution as the reference. AGI computes class-specific adversarial examples as references. For the experiments on the ImageNet-2012 dataset (Russakovsky et al., 2015), we select 10 references and 5 interpolations (k=5) for IG-Uniform, IG-SG and IG-SQ. Besides, we chose 50 references and one random interpolation (k=1) for EG, and set 10 interpolations (k=10) and 5 class-specific adversarial references for AGI as the recommended settings in these papers. For a fair comparison, we ensure all these methods use the same number of 50 back propagations. Considering the small input size of CIFAR-10 and CIFAR-100 datasets (Krizhevsky et al., 2009), we employed 30 back propagations for all the baseline attribution methods. We employed the same hyperparameter settings for all the experiments in this paper (e.g., with pixel perturbation, DiffROAR, Sensitivity-n, etc.). In addition to the baseline methods, other compared methods that rely on multiple gradient computations including Integrated Gradients (IG) and SmoothGrad (SG) are also implemented with the same hyperparameters. In IG, we produce 50 interpolations (k=50) for the final integral estimation. In SG, we average 50 gradients on inputs with Gaussian noise.

**Hyperparameter Choice of Explained Models.** Three deep models including VGG-16, ResNet-34 and PreActResNet-18 are chosen to be explained by different attribution methods in our experiments. For the pixel perturbation and Sensitivity-n benchmarks, VGG-16 and ResNet-34 networks are trained on the ImageNet-2012 training dataset. For the DiffROAR benchmark, we first train a PreActResNet-18 on CIFAR-10 and CIFAR-100 training datasets separately. Then, the PreActResNet-18 is fine-tuned for a total of 10 epochs with the initial learning rate $10^{-2}$ decayed by 10 on the 5-th and 7-th epochs.

### A.3.2 EXPERIMENTAL SOFTWARE AND PLATFORM

All the experiments were conducted on a Linux machine with an NVIDIA GTX 3090Ti GPU with 24GB memory and a 16-core 3.9GHz Intel Core i9-12900K CPU and 125GB main memory. All attribution methods are tested and trained on PyTorch deep learning framework (v1.12.1) with Python language.

## A.4 EXTENDED EXPERIMENTS

Below extended experiments are conducted to test the efficacy of the proposed technique.

### A.4.1 EXTENDED EXPERIMENTS ON VALID INTERPOLATIONS AND VALID REFERENCES

In this paper, we propose two techniques to identify valid interpolations and references for estimating an integral along a non-linear integration path. Here, we provide more detailed experiments for further understanding the valid interpolations and references. We randomly sample 1,000 testing images from the ImageNet 2012 Validation set to show the effectiveness of both valid interpolations and references. Here, we take IG-Uniform as the baseline attribution method. On both ResNet-34 and VGG-16, we re-calibrate attribution maps produced by IG-Uniform with valid interpolations and references. In Figure 7 and 8, we show the AUC of the fractional output change difference under different numbers of references with a fixed number of interpolations (k=5).

**Both valid references and interpolations can re-calibrate attributions.** In comparison with the experimental results of the vanilla attribution map produced by IG-Uniform in Figure 3, Figure 7 and Figure 8 show that both valid interpolations and references can improve the interpretability of IG-Uniform by a large margin. The significant improvement demonstrates the effectiveness of the assumed desired reference and the proposed re-calibration technique.

**Valid References lead to efficient integral estimation.** Different from valid interpolations, valid references employ importance sampling to efficiently estimate the integral. Figure 7a shows that integrating with valid references enables the attribution method to achieve high performance with

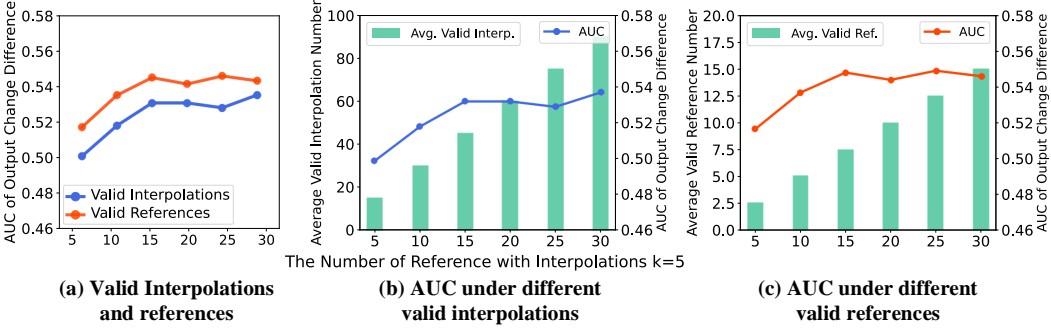

(a) Valid Interpolations
and references

(b) AUC under different
valid interpolations

(c) AUC under different
valid references

Figure 7: The comparison between valid interpolations and valid references on ResNet-34. **(a)** The AUC of fractional output change difference comparison between valid interpolations and references. **(b)** The relationship between the AUC and average valid interpolations. **(c)** The relationship between the AUC and average valid references.

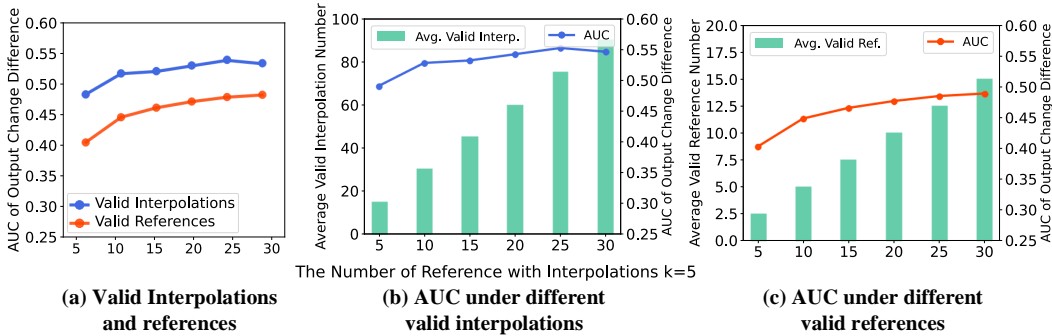

(a) Valid Interpolations
and references

(b) AUC under different
valid interpolations

(c) AUC under different
valid references

Figure 8: The comparison between valid interpolations and valid references on VGG-16. **(a)** The AUC of fractional output change difference comparison between valid interpolations and references. **(b)** The relationship between the AUC and average valid interpolations. **(c)** The relationship between the AUC and average valid references.

a small number of references. However, estimating an accurate integral with valid interpolations requires a large number of references.

**More references lead to high performance.** Figures 7b&7c and Figure 8b&8c shows the relationship between the AUC of fractional output change difference and the number of references. We can observe that more references and interpolations can lead to high performance by re-calibrating with both valid interpolations and references, which demonstrates that more references can cover a more accurate integration path.

**Drawbacks and superiority of valid references.** In Figure 8, we can observe that IG-Uniform re-calibrated with valid references cannot outperform the results calibrated with valid interpolations on VGG-16, which is inconsistent with the results on ResNet-34. The reason is that the valid reference relies on a relaxed condition to approach an accurate integral. However, VGG-16 has a more complicated decision pathway, which is hard to estimate an accurate integral along a linear integration path. As compared to VGG-16, ResNet-34 with a more flattened decision pathway enables attribution methods to obtain accurate results with valid references, which also aligns with the existing works that show the residual block can flatten the loss landscape (Li et al., 2018). Although a complicated model impedes the performance of valid references, valid references can still improve the reliability of baseline methods. Moreover, identifying valid references require less memory consumption than identifying valid interpolations. Thus, the efficiency of identifying valid references enables the attribution method incorporated into the model training process. In addition, training a robust model can flatten the loss landscape of the model, which also favors the proposed valid references for robust model training. In Appendix A.4.2, we perform experiments of regularizing the behavior of the model with feature attribution.

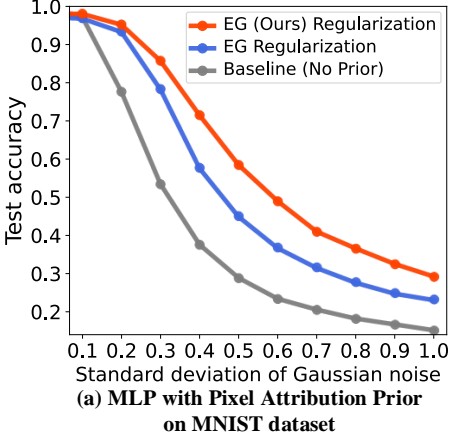 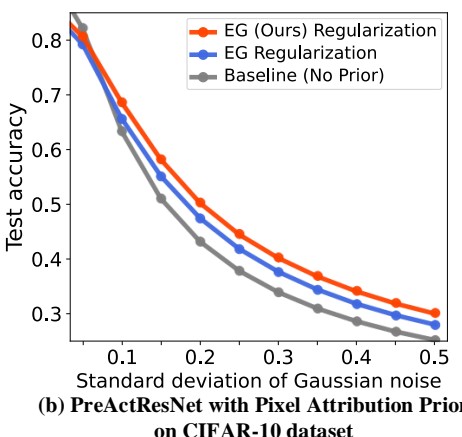

(a) MLP with Pixel Attribution Prior on MNIST dataset   (b) PreActResNet with Pixel Attribution Prior on CIFAR-10 dataset

Figure 9: The accuracy comparison under testing images with the increased standard deviation of Gaussian noise between an original model, the model trained with EG regularization term, and the model trained with EG (Ours) regularization term. **(a)** The accuracy comparison of three MLP models trained on MNIST dataset. **(b)** The accuracy comparison of three PreActResNet models trained on MNIST dataset. **Higher** values indicate **better** results.

### A.4.2   EXTENDED EXPERIMENTS ON ATTRIBUTION PRIOR

In addition to explaining a model prediction, attribution methods are also expected to regularize the behavior for constructing a robust model. Therefore, many regularization terms (Ross et al., 2017; Erion et al., 2021) are proposed to regularize the behavior by the attributions for obtaining a robust model. Erion et al. (2021) assert a prior for the image classification task that adjacent features or pixels should have similar attributions for the model prediction. To regularize the model behavior with this prior knowledge, they employed a Laplace zero-mean prior on the difference between attributions of neighborhood pixels to encourage the adjacent pixels to have similar attributions. Since the regularization term encourages the input pixels to make the prediction jointly, the trained model is robust to the input with Gaussian noise. In this part, we also combine the prior with the re-calibrated attributions as the regularization term for training a robust model. We train two robust models with the regularization term on MNIST (LeCun et al., 1998) and CIFAR-10 (Krizhevsky et al., 2009) datasets. For MNIST, we train a multi-layer perceptron (MLP) with two convolution layers and one fully connected layer. On CIFAR-10 dataset, we take a PreActResNet-18 (He et al., 2016a) as the baseline network. On both datasets, we employ one interpolation (k=1) and 5 references randomly sampled from the training dataset in the training process. For each dataset, we compare the original model, the model trained with EG regularization and the model trained with re-calibrated EG regularization to test their robustness on the test set with Gaussian noise. Figure 9 shows the accuracy change with the increase of the standard deviation of Gaussian noise on the three models. Although two robust models (EG and EG Ours) can improve the robustness of the baseline model on Gaussian noised input, we can observe that our method can outperform the regularization term with Expected Gradients (EG) on both two datasets, which demonstrates that the proposed re-calibration technique can effectively enhance the reliability of EG.

### A.4.3   EXTENDED EXPERIMENTS ON SENSITIVITY-n

In this part, we provide more experimental results on Sensitivity-n metric. Figure 10 shows the results of ResNet-34 on Sensitivity-n, which shows that the proposed technique can also improve the sensitivity of the model on ResNet-34. In addition, Figure 11 shows the results of PreActResNet-18 on CIFAR-10 and CIFAR-100 datasets on Sensitivity-n metric. These experimental results also ascertain that our re-calibration technique enables the vanilla methods to improve their sensitivity under smaller sampling set sizes. Since Sensitivity-n benchmark randomly removes pixels from the input to test their sensitivity, the re-calibrated attribution maps with well-aligned explanations for the foreground object are more robust to the random removal than the vanilla attribution maps.

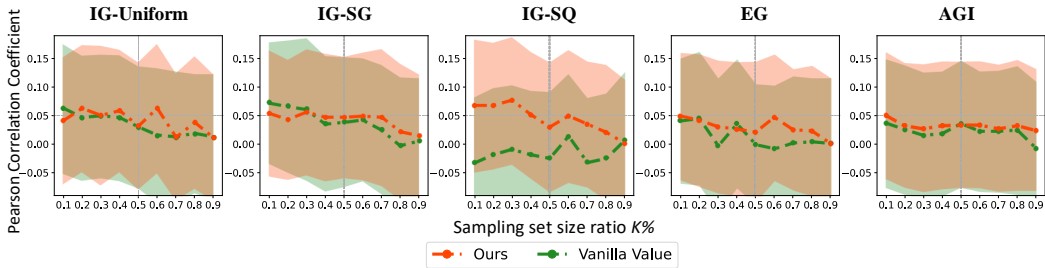

Figure 10: Experimental results of ResNet-34 on Sensitivity-n benchmark. **Higher** values indicate **better** results.

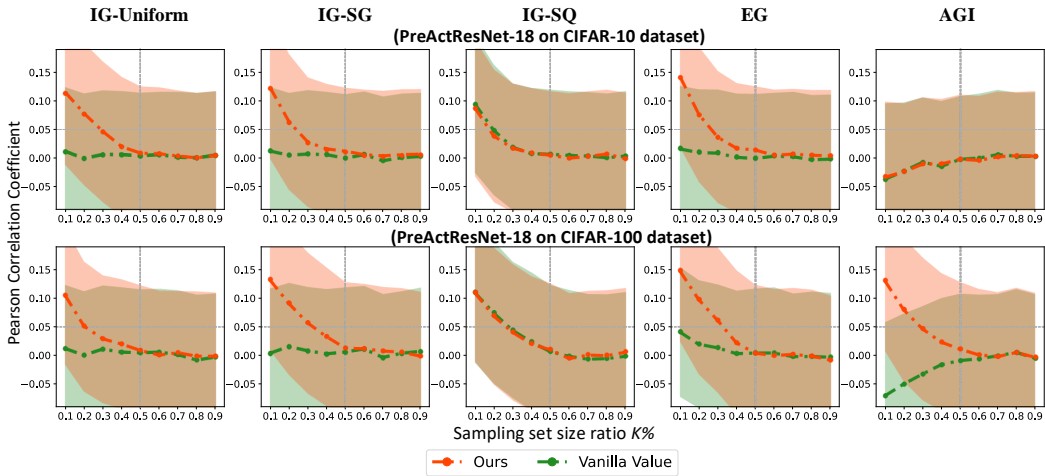

Figure 11: Experimental results of CIFAR-10 and CIFAR-100 datasets on Sensitivity-n benchmark. **Higher** values indicate **better** results.

### A.4.4 EXTENDED EXPERIMENTS ON SANITY CHECKS

In addition to the quantitative evaluation benchmarks, Adebayo et al. (2018) proposed two sanity checks that attribution methods should pass including: (i) the method should provide different attributions for the model trained with randomly permuted labels, and (ii) the attribution map should show a substantial change to the model parameter randomization. In this part, we test the baseline methods and their re-calibrated variants for the two sanity checks.

**Data randomization test.** Since deep models can easily fit random labels Zhang et al. (2017), data randomization test permutes the training labels to train a model and test the sanity of attribution methods. In Figure 12, we compare the attribution methods for the models trained with true and random labels. The first four columns show the comparison between the attribution maps calibrated with our technique and taken absolute values. The second four columns show the comparison between re-calibrated attribution maps and the vanilla attribution maps. Compared with the baseline methods, our method leads to a larger change in the calculated attribution maps. In addition, we quantitatively compare the similarity between the attribution maps trained with true and random labels. Figure 13 shows the comparison of Spearman rank correlation between the attribution maps produced by the baseline methods and re-calibrated methods. Compared with both absolute attributions and vanilla attributions, the experimental results show that the proposed re-calibration technique improves the sanity of the baseline attribution methods for the data randomization test.

**Model parameter randomization test** In addition to the sanity of the data randomization, the attribution methods are also expected to pass the model parameter randomization test. In our experiments, we successively randomize the weights of an Inception V3 model (Szegedy et al., 2016) from top to bottom layers. Figures 14 - 17 show the examples of the attributions maps as the parameter cascading from the top to bottom. We mark attribution maps produced by the baseline method

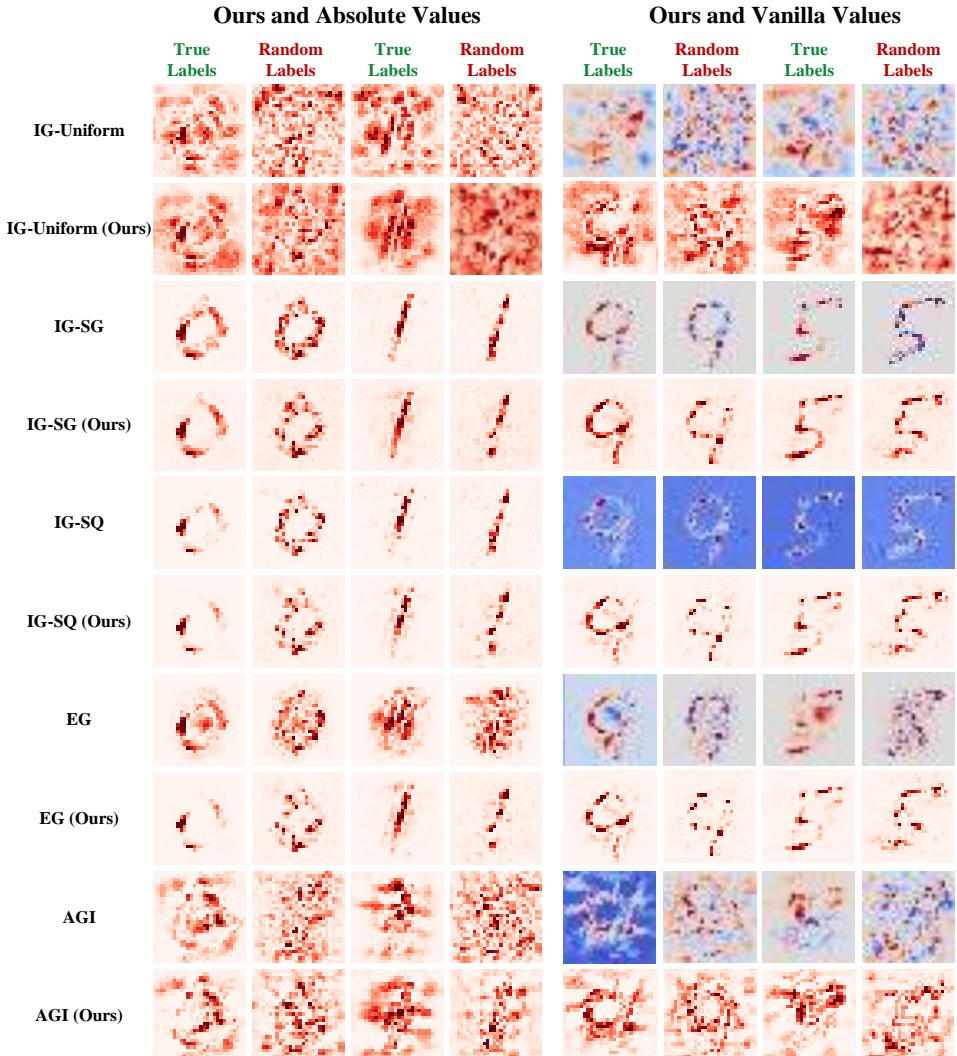

Figure 12: Data randomization test for the baseline attribution maps (vanilla and absolute values) and the proposed re-calibrated attribution maps for a true label trained model and a random label trained model.

and our re-calibration method with blue boxes for easy comparison. We can observe that the calibrated attribution maps are more sensitive with the increased randomization layer numbers, which shows the proposed technique enables the baseline methods to improve the sensitivity for the model parameter randomization test.

## A.5    VISUAL INSPECTION

In this section, we provide visualization examples of calculated attribution maps for visual inspection. Figures 18 - 20 show the comparison of vanilla attribution maps, attribution maps computed with absolute values and the attribution maps re-calibrated with our technique on ResNet-34. In addition, Figures 21 - 23 show the comparison of attribution maps on VGG-16. We can observe that the attributions re-calibrated with our technique enable the maps to better align with the foreground objects. In addition, vanilla attribution maps suffer from a large variation between positive and negative scores on both ResNet-34 and VGG-16.

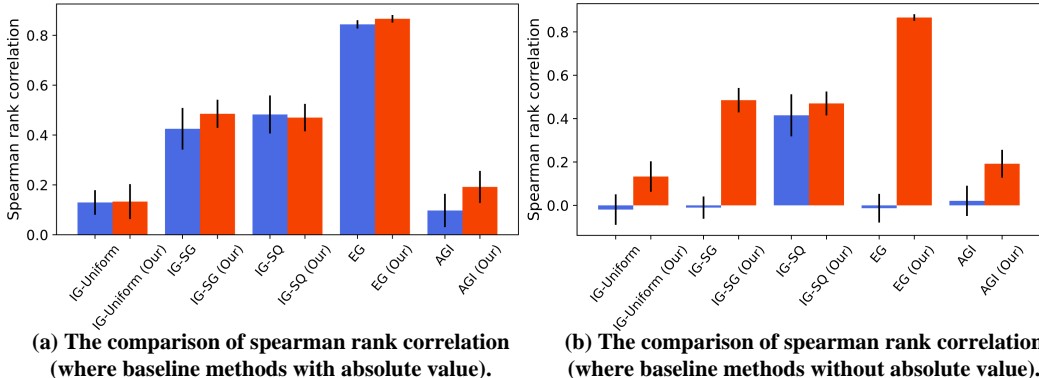

**(a) The comparison of spearman rank correlation (where baseline methods with absolute value).**

**(b) The comparison of spearman rank correlation (where baseline methods without absolute value).**

Figure 13: Quantitative evaluations for data randomization test. **(a)** The spearman rank correlation for baseline attribution maps with absolute values and re-calibrated attribution maps. **(b)** The spearman rank correlation for baseline attribution maps without vanilla values and re-calibrated attribution maps. **Larger** correlations indicate **better** results.

## A.6    LIMITATIONS

In this paper, to improve the inconsistency of integral-based attribution methods, we assume different references for different input features to re-calibrate attributions. Although our method has been shown to enhance the reliability of popular attribution methods, the modification also carries the risk of violating the completeness axiom. Therefore, exploring an explainable method that guarantees the consistency between practice and their theoretical guarantee and their practical implementation is still an important task. In addition, we devise our method based on theoretical insights. However, precisely quantifying the impact of these insights is not possible due to the intrinsic ambiguities in the quantitative evaluation metrics used in this research direction. This is a limitation of this domain in general. Nevertheless, we verify that our method consistently improves the evaluation scores. In practice, it may be important to verify that the used evaluation metric represents the real-world scenario appropriately.

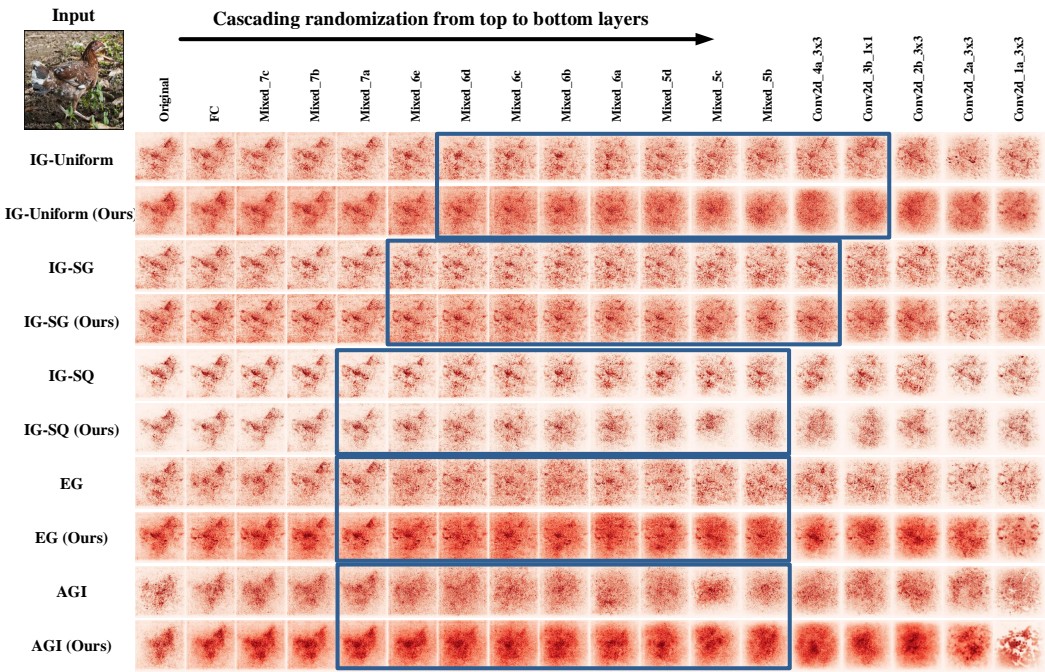

Figure 14: Model parameter randomization test on Inception v3 for the testing image from the ImageNet-2012 dataset. The attribution maps produced by baseline methods and our calibration method are compared as cascading randomization from top to bottom layers.

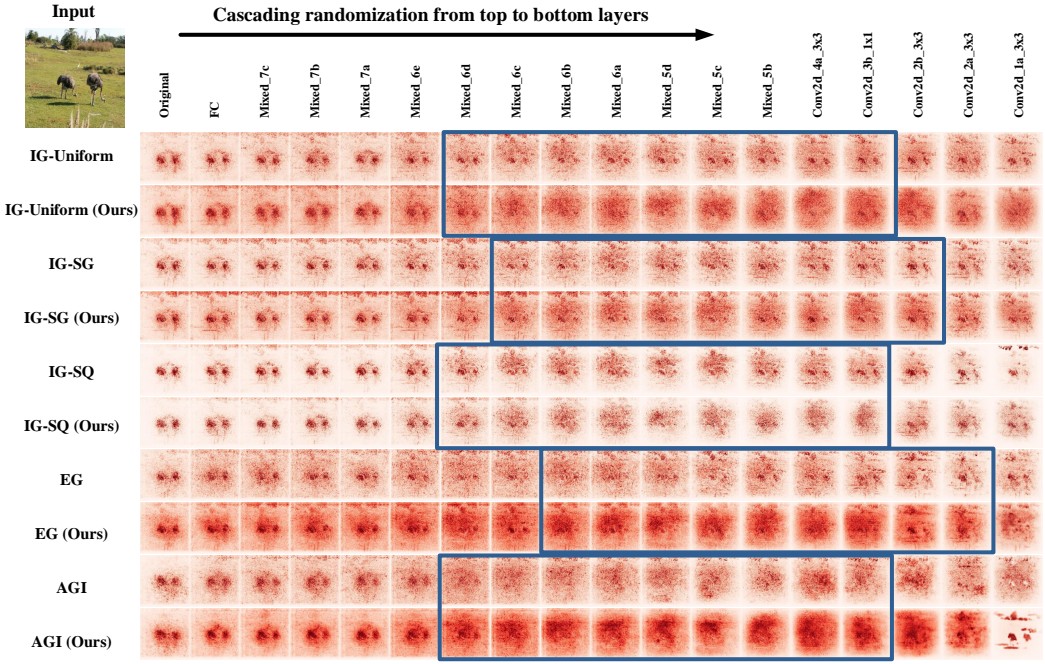

Figure 15: Model parameter randomization test on Inception v3 for the testing image from the ImageNet-2012 dataset. The attribution maps produced by baseline methods and our calibration method are compared as cascading randomization from top to bottom layers.

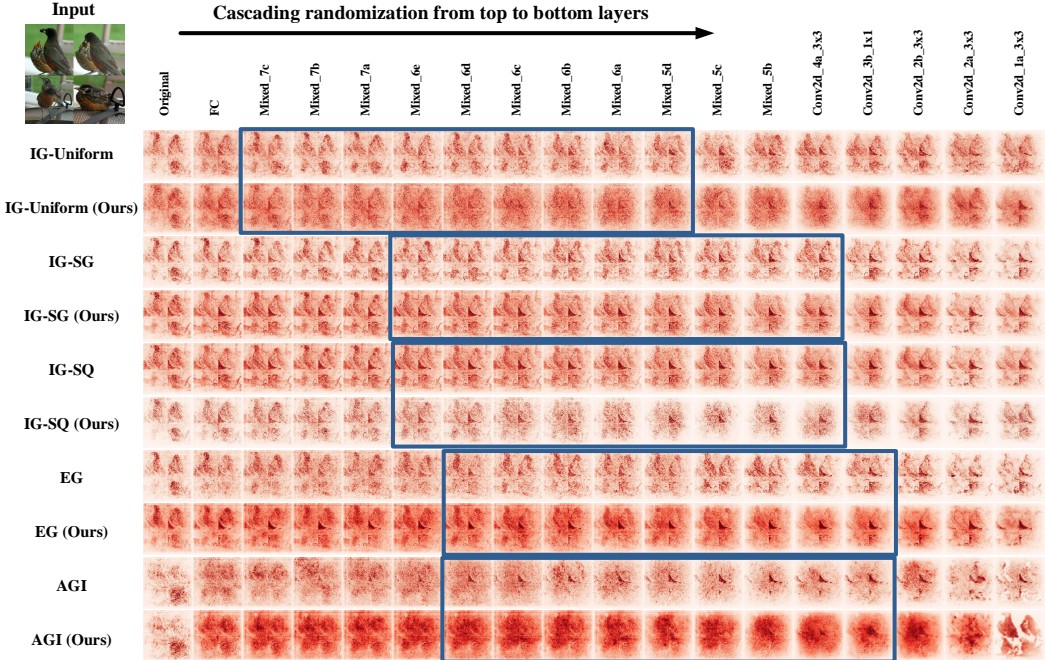

Figure 16: Model parameter randomization test on Inception v3 for the testing image from the ImageNet-2012 dataset. The attribution maps produced by baseline methods and our calibration method are compared as cascading randomization from top to bottom layers.

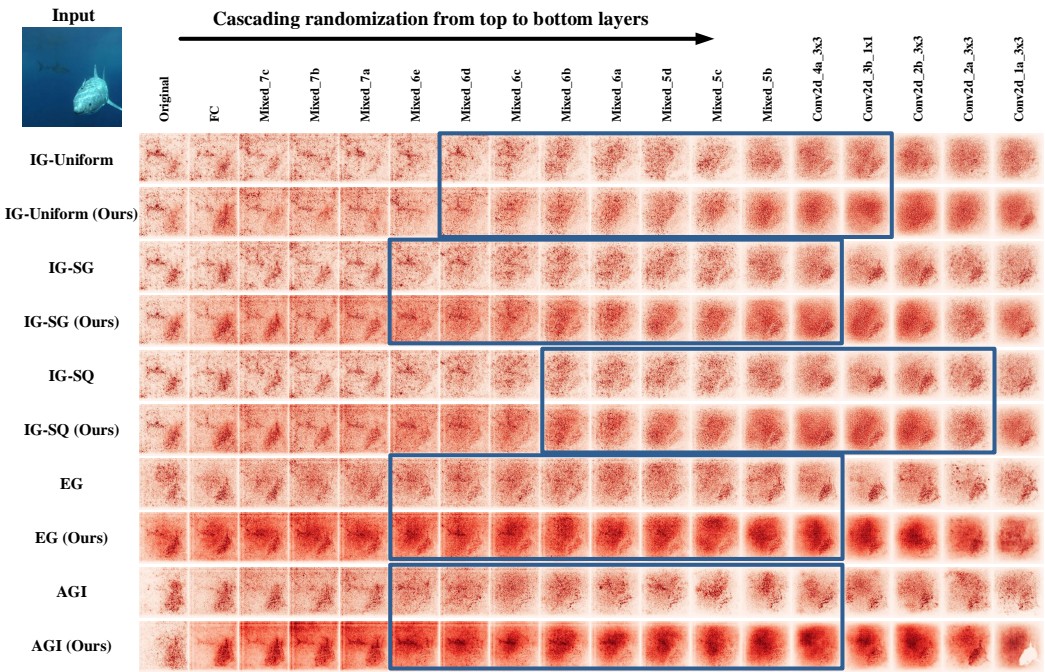

Figure 17: Model parameter randomization test on Inception v3 for the testing image from the ImageNet-2012 dataset. The attribution maps produced by baseline methods and our calibration method are compared as cascading randomization from top to bottom layers.

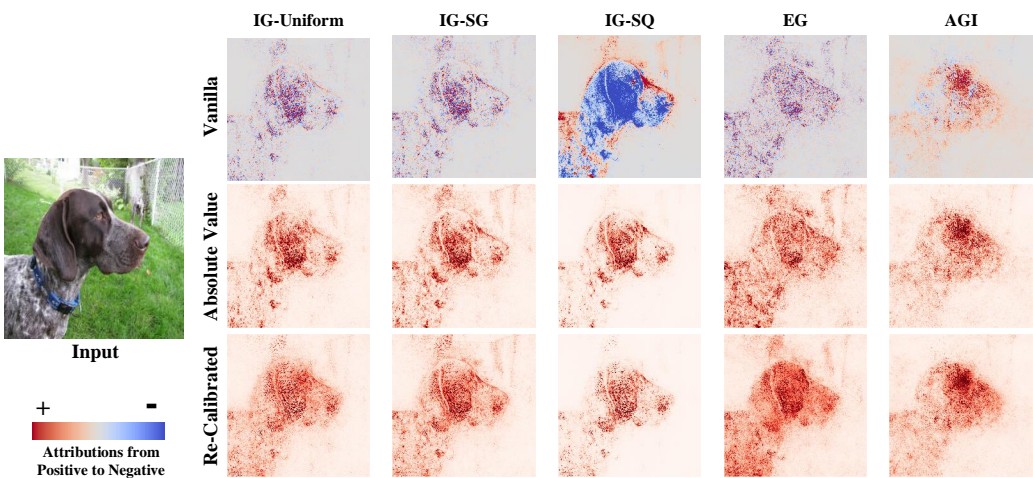

Figure 18: The comparison of vanilla attribution maps, attribution maps taken absolute values, and attribution maps calibrated with the re-calibrated technique on ResNet-34.

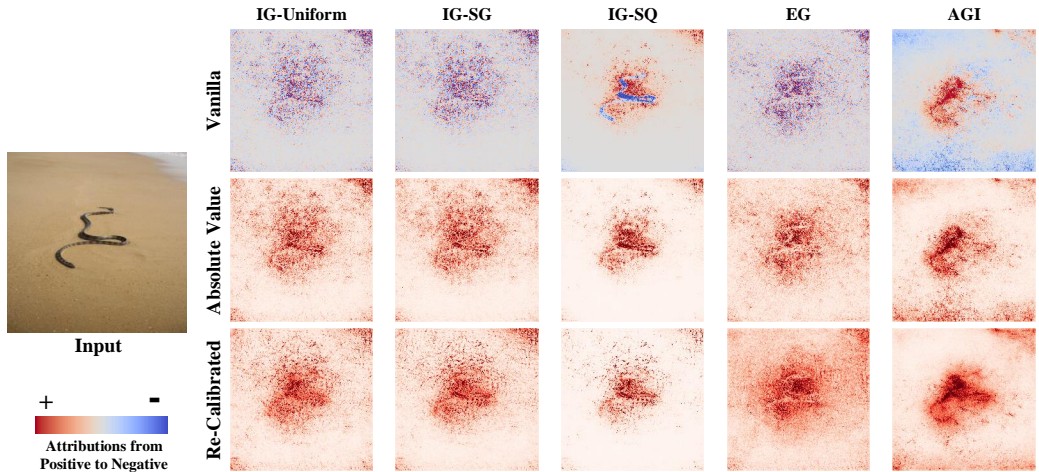

Figure 19: The comparison of vanilla attribution maps, attribution maps taken absolute values, and attribution maps calibrated with the re-calibrated technique on ResNet-34.

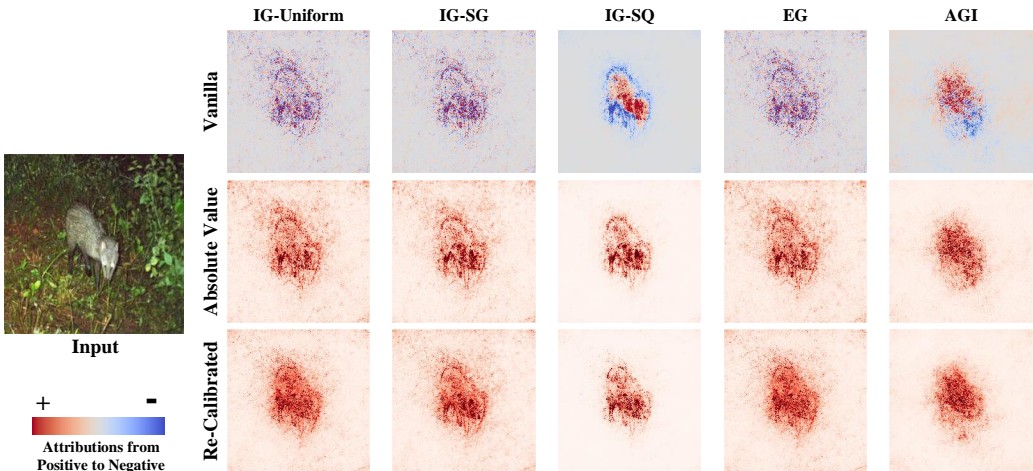

Figure 20: The comparison of vanilla attribution maps, attribution maps taken absolute values, and attribution maps calibrated with the re-calibrated technique on ResNet-34.

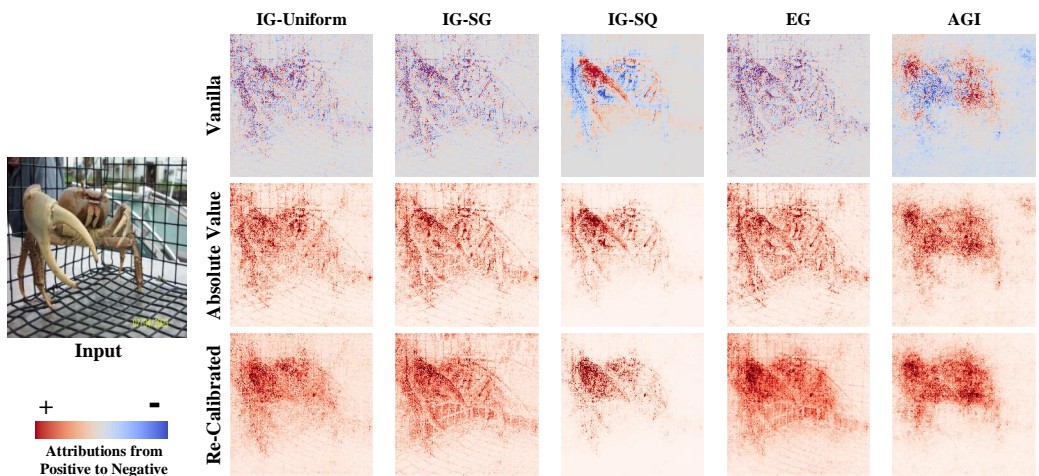

Figure 21: The comparison of vanilla attribution maps, attribution maps taken absolute values, and attribution maps calibrated with the re-calibrated technique on VGG-16.

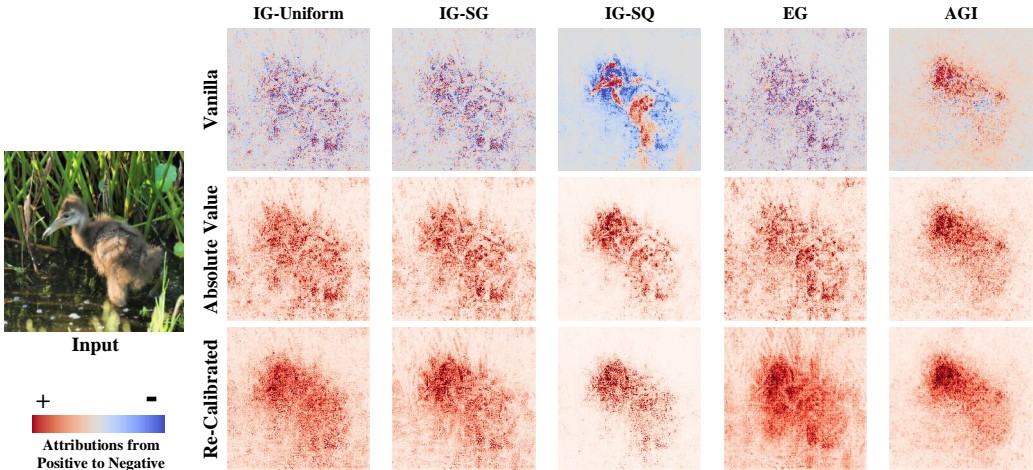

Figure 22: The comparison of vanilla attribution maps, attribution maps taken absolute values, and attribution maps calibrated with the re-calibrated technique on VGG-16.

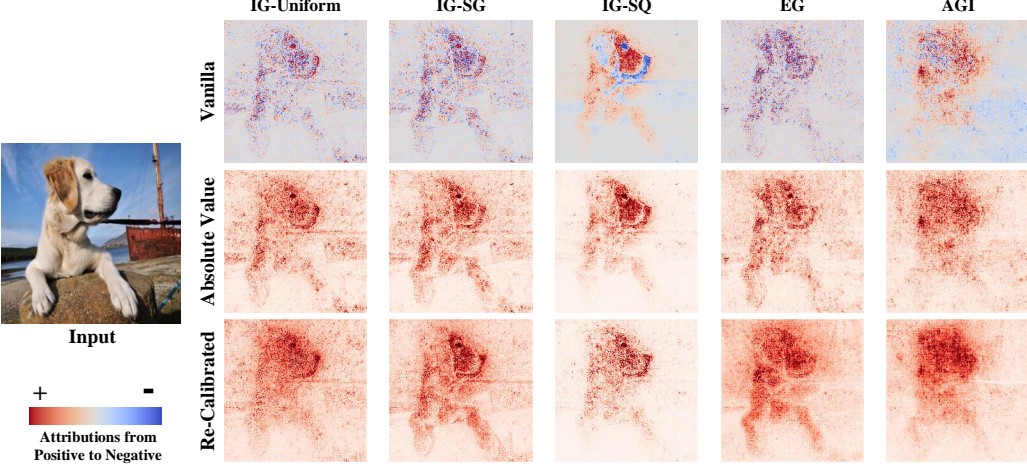

Figure 23: The comparison of vanilla attribution maps, attribution maps taken absolute values, and attribution maps calibrated with the re-calibrated technique on VGG-16.

