# OpenReview forum: "Re-calibrating Feature Attributions for Model Interpretation"
_ICLR.cc/2023/Conference — ICLR 2023 notable top 25%_

### Official Review · Reviewer_rkn8 · 2022-10-23

**Confidence:** 3
**Correctness:** 3
**Technical Novelty And Significance:** 4
**Empirical Novelty And Significance:** 3
**Recommendation:** 8

**Clarity, Quality, Novelty And Reproducibility:**

**Clarify**

If taking a very deep look, the main body of this paper is clear but it is still generally reader unfriendly. I would suggest that the author improve its writing (with suggestions mentioned above) to not puzzle readers especially for those who are not exactly working in this area/technical direction (but still want to know more here given this area is quite important).

**Quality and Novelty**

Technically novel enough, with several good insights like designing an appropriate "reference" instead of relying on manual design, finding the good interpolation points for path integral and an efficient sampling methods.

**Reproducibility**

Good given source code is provided.



**Strength And Weaknesses:**

**Strength**

1. This paper brings a lot of meaningful technical depth. It deeply analyzes a tricky issue of the previously widely used method, and proposed the solution based on detailed mathematical analysis. A little bit surprisingly that the finally derived method seems to be simple with such a non-trivial technical setup which is impressive.

2.  It is also technically sound and rigorous, with a) a good mathematical system; b) solid empirical verifications of the method.

**Weakness**

1. Mostly this paper is not reader friendly as the writing seems too technically heavy. In particular, a) a lot of long-sentences are used which should be further broken down; b) some concepts are introduced with a little bit sudden manner, especially in Abstract (e.g., "attribution scores", "reference" and "local and global evaluation metrics").

**Summary Of The Paper:**

This paper improves the previous path integration based model interpretation works by overcoming the limit of absolute based attribute scores. The method is mainly consisting of a) building a better "reference" than the manual way previously; b) identifying the interpolation point in the non-linear path through the relationship of "variation" and gradient; c) an efficient sampling method to make the computation tractable. The final method looks simple with a "re-calibratable" property on top of the current integral-based methods.

**Summary Of The Review:**

Overall I think this is a good paper especially for the particular area of path integration based model understanding/feature attribution. It solves a deep and tricky limitation of previous methods with enough technical novelty/depth for ICLR publications. I would vote for its acceptance conditioned on a good re-writing of the paper: currently it is presented in a too technically heavy way which is not good for the broader ICLR committee.

---

> ### Author Response · Authors · 2022-11-14
> **Response to rkn8**
>
> We are grateful for your time and constructive feedback. We will improve the camera ready version considering your input.
>
> > Mostly this paper is not…especially in Abstract (e.g., “attribution scores”, “reference” and “local and global evaluation metrics”).
>
> In the camera-ready version, we will improve the paper along the aspects mentioned by the reviewer. In particular, we will break the long sentences into shorter ones. Also, we will ensure that the technical terms are not used in the text before the explanation of the concept. This requires minor changes in the text.
>
> > If taking…. Is quite important).
>
> We thank the reviewer for the feedback, we will improve the camera-ready version as indicated above.
>
>
> > Overall I think… broader ICLR committee.
>
> We are grateful for this very relevant comment that can help improve the impact of our paper. We will ensure improving the camera-ready version keeping in view the broader ICLR community.

---

### Official Review · Reviewer_FTMa · 2022-10-25

**Confidence:** 4
**Correctness:** 3
**Technical Novelty And Significance:** 3
**Empirical Novelty And Significance:** 3
**Recommendation:** 8

**Clarity, Quality, Novelty And Reproducibility:**

The paper is clear and logically structured. The authors suggest the code for reproducibility is in the supplementary material, but I could only find the experimental setup description, making it difficult to reproduce the experiments.
The novelty is incremental over existing techniques, but the results are promising. The analysis is not new but clear and consistent. I was expecting more insights from the authors.


**Details Of Ethics Concerns:**

I have no ethical concerns about the paper.


**Strength And Weaknesses:**

+ The proposed approach is interesting and presents promising results.
+ The idea of calibrating attribution scores to capture relevant features and interactions is promising but has been around for a while. The novelty is incremental and helps solve some limitations of the IG approach.
+ The experiments and analysis are clear, but the discussion could be improved. The authors have the opportunity of creating a link between uncertainty and explainability by exploring more about the relationship between calibration and interpretation.
+ The integrated gradients method can assign different attributions to features that have the same effect on the model or assign positive attributions to features with no effect. Does the proposed approach solve those problems? I miss some qualitative discussions about the method performance.
- The authors did not discuss the limitation of the proposed method. Therefore, it will be meaningful to discuss the gap between the experiments in the current version of the paper and the real-world applications.


**Summary Of The Paper:**

This paper proposes a method to create the reference image for Integrated Gradient attribution approaches to interpret model predictions. The method re-calibrates the attribution scores without the additional computational overhead of the traditional approaches. The authors suggest this strategy is more relevant to model prediction and feature interpretation. The paper considers different aspects of the attribution techniques during the evaluation and obtains promising results.

**Summary Of The Review:**

The proposed paper presents a promising approach with interesting results and analysis. The discussion is direct, but I was expecting more insights. The authors also should discuss the proposed approach's limitations.

---

> ### Author Response · Authors · 2022-11-14
> **Response to FTMa**
>
> We are grateful for your time and constructive feedback.
>
> > The idea… The novelty is incremental and helps solve some limitations of the IG approach.
>
> We enhance the path attribution framework (as a whole, not just IG) in a principled manner, such that multiple existing methods can benefit from our new insights. In our opinion, the magnitude of our novelty should be judged from the viewpoint of resolving fundamental issues with the path attribution framework. In addition to being theoretically motivated, our contribution promises a clear positive impact on the field.
>
>
> > The experiments…The authors have the opportunity of creating a link between uncertainty and explainability by exploring more about the relationship between calibration and interpretation.
>
> We thank the reviewer for the suggestion. In fact, we did provide results on using our method for attribution prior in Section A.3.2 in the supplementary material, which relates to model explainability. Attribution prior is used to regularize models for robustness, that has a direct relation with the interpretability of the model. We show that our method is able to positively impact this downstream task. We leave further exploration in this direction for future work.
>
>
> > The integrated gradients… Does the proposed approach solve those problems?
>
> The integrated gradients (IG) method is an axiomatic approach, yet it misbehaves. We believe the reason behind this phenomenon is the existence of invalid features on the integration path. This is exactly what our method addresses. As per theoretical grounds, if IG ensures that the integration is done with appropriate reference and path features, it should not misbehave. This is hard to verify quantitatively because the evaluation metrics in this domain are generally not considered fully reliable themselves. However, we do see a consistent (considerable) gain in IG performance with all baselines under our enhancement. This indicates that we are able to fix the fundamental issues with IG.
>
>
> > The authors did not discuss the limitation… and the real-world applications.
>
> We devise our method based on theoretical insights. However, precisely quantifying the impact of these insights is not possible due to the intrinsic ambiguities in the quantitative evaluation metrics used in this research direction. This is a limitation of this domain in general. Nevertheless, we verify that our method consistently improves the evaluation scores. In practice, it may be important to verify that the used evaluation metric represents the real-world scenario appropriately. We will mention this.

---

### Official Review · Reviewer_brex · 2022-11-02

**Confidence:** 3
**Correctness:** 4
**Technical Novelty And Significance:** 2
**Empirical Novelty And Significance:** 2
**Recommendation:** 5

**Clarity, Quality, Novelty And Reproducibility:**

For details, please see strengths & weaknesses above.

In its current form, the submission lacks clarity, which makes the quality (apart from quantitative improvements in the metrics) difficult to judge. If I understand the proposed approach correctly (average over set of ReLU-activated IG attributions), the originality is limited. The results seem easily reproducible with the experimental description as provided by the authors.

**Strength And Weaknesses:**

### Strengths
- The experimental evaluation is extensive and shows that the proposed approach yields consistent improvements over a wide range of different settings, as measured by various metrics for assessing the quality of attribution methods.
- The authors aim to address an important shortcoming of commonly employed integral-based attribution methods, namely their dependence on the chosen baselines as well as an inconsistency between the theoretical motivation for IG methods and their practical usage.
- The paper is generally well structured and (apart from sections 4.2 and 4.3, see weaknesses) very well written and easy to follow.

### Weaknesses
While the experimental results show promising gains in the employed metrics for interpretability and the approach is well motivated, I am still hesitant to recommend the paper for publication in its current form. If the authors are able to address my concerns, I am open to increasing my score.

- **Lack of clarity w.r.t. method and motivation**: I have the following issues with and open questions regarding the method and would appreciate if the authors could elaborate.
	- As far as I understand, the proposed method is motivated (apart from the described *inconsistency*) by the high spatial variance in the attribution maps of integral-based methods, as shown in Fig. 1 (top row). Commonly, the absolute value is taken to overcome this problem (see Fig. 1, second row), which the authors criticise as yielding attribution maps that are inconsistent with the underlying theory and thus do not faithfully reflect the explained model. Judging from Algorithm 1, the main difference of the proposed method is that the authors employ a ReLU activation over the pixel attributions for each reference image and average the result — *i.e.* instead of taking the absolute value to 'hide' the high variance in the attribution maps, the authors simply clamp the attributions at zero to do so. As such, while the experimental results seem to speak in favour of the proposed method, I currently fail to see how this approach is fundamentally different w.r.t. being better motivated than taking the absolute value.
	- There is a stark contrast between the simplicity of the actual implementation of the proposed method (Algorithm 1) and its description and derivation. To exaggerate a bit, it seems like a very complicated way of saying "Instead of taking the absolute value, we average the ReLU-activated attributions over multiple reference images". This is aggravated by a lack of clarity in the method description (see next bullet).
	- After carefully reading the method section multiple times, I am still confused as to how the reference set D is obtained. In Algorithm 1, the reference set seems to be given, while in the text it says the reference set is "obtained by the method's own strategy". As far as I can tell, this seems not to be referring to the authors' method, but simply to the reference set as used in prior work (e.g., uniform noise), i.e. in the respective base method which the authors extend by their approach. However, this seems to be contradicting the statement that "we \[...\] modify the input image with the model gradients to construct the reference", as well as the general notion of "\[computing\] the reference along the gradient ascending direction \[...\] \[similar to\] Adversarial Gradient Integration", which is also centrally placed in the abstract ("The reference is computed in a gradient ascending direction on the model's loss surface"). I would highly appreciate if the authors could clarify.
	- What exactly is the relevance of Lemma 1? What is different from standard Integrated Gradients, which also estimates the integral in eq. 3?
	- The mathematical notation / naming is not optimal, which makes the method section difficult to read:
		- In eq. (3) the index i is used to index a specific feature dimension of an input x, but also to index a specific reference sample in the set D.
		- The equivalence between x' and D_i is also confusing—why not simply use D_i (optimally with a different index)? Further, using the term *variation* for the *difference* between two points is confusing.
		- What exactly is the set of signed input gradients in the second paragraph of 4.2.? What does this notation mean exactly (similarly for the variation / difference)?

- **Missing reference**: Guided Integrated Gradients by Kapishnikov et al. (CVPR 2021) seems to be a highly relevant prior work that should be included in the discussion of related work as well as in the experimental evaluation. What are the main differences to and advantages over Guided Integrated Gradients?

Additional minor issues:
- The result graphs are difficult to parse. I would recommend to additionally summarise the curves via a single number per method (e.g., AUC). This concerns all result Figs.
- Further, the relevant comparisons (e.g., ResNet (Ours) vs. ResNet (Baseline)) are not easy to make in the current graphs, I would recommend grouping the results for better comparability. This is particularly difficult in Fig. 4—here, the relevant baselines are seemingly not even included (e.g., IG-SQ* is shown, but not IG-SQ etc.)?
- In Fig. 6, the shaded areas are not explained in the caption, what do they represent?

**Summary Of The Paper:**

The authors propose a method to improve the path along which the gradients are integrated in integral-based attribution methods for explaining the decisions of deep neural networks (DNNs).

This method is based on the observation that while integrated gradients (IG) are theoretically well motivated, the necessary assumptions are often violated in common practice (e.g., by taking the absolute value of the computed attributions). To overcome this, the authors propose a method to select "valid" interpolation points for the integration, which are independently chosen for each input feature.

In a wide range of experiments with different datasets (CIFAR-10, CIFAR-100, ImageNet), the authors report consistent improvements according to various metrics (pixel perturbation, DiffROAR, sensitivity-n) when extending commonly used integral-based methods (IG-Uniform, IG-SG, IG-SQ, EG, AGI) with their proposed method for various networks (VGGs and ResNets).

**Summary Of The Review:**

While the experimental results show promising gains in the employed metrics for interpretability, I am hesitant to recommend the paper for publication in its current form. If the authors are able to address my concerns, I am open to increasing my score (see weaknesses).

---

> ### Author Response · Authors · 2022-11-14
> **Response to brex (1/2)**
>
> We are very grateful for the thorough review and constructive comments. We find that the raised concerns are mainly due to misunderstandings. We clarify them below and will incorporate clarifications in the camera ready version.
>
> > “As far as…..I currently fail to see how this approach is fundamentally different w.r.t. being better motivated than taking the absolute value.
>
> > There is a stark contrast between the simplicity of the actual implementation of the proposed method (Algorithm 1) and its description and derivation…..”
>
> Our approach is fundamentally different from the existing practice. We do not use any heuristic to get rid of the spatial variance (with ReLU or absolute values). The spatial variance problem automatically gets resolved in our approach in a principled manner. We introduce a key notion of ‘valid’ interpolation points. These points have the property that they fall on the paths which are along the gradient ascending directions w.r.t. the input. We systematically identify those points by analysing their values, and integrate the gradients using only the valid interpolation points for each pixel. Our method allows the use of multiple references and treats each pixel individually in identifying the valid interpolations. For tractability, we employ importance sampling. Algorithm 1 is a pseudo-code that abstracts away unnecessary theoretical details. We do not use ReLU, and the averaging follows naturally from the importance sampling. Algorithm 1 is only a part of our contribution, and should be treated as such. It is provided as a blue-print to readily re-calibrate the existing methods with our findings.
>
> For a high-level understanding: While integrating the gradients, existing methods end up using inappropriate points on the path. This leads to high spatial variance in attributions, which previous methods fix in a heuristic manner, thereby contradicting the theory on which the path attribution scheme and the evaluation metrics are built. We have presented a technique that fixes the issue in a principled manner.
>
> > After carefully….The reference set seems not to be referring to the authors' method, but simply to the reference set as used in prior work.
>
> Thank you for this important query. When we say “method’s own strategy”, we mean the strategy of the method being re-calibrated (e.g., if we are recalibrating IG, then $D$ is the set that contains the reference image originally proposed by IG). However, we do not need to explicitly replace the reference set $D$ (for any method) with our idea of “reference in the gradient ascending direction”.  This is because, when identifying the valid interpolations, we also consider the points present in the set $D$. If those points are indeed in the gradient ascending directions, then they get used unaltered. If they are not, then they effectively get altered to the interpolated points that satisfy this property. In that case, the path on which we integrate will automatically not include the original reference. Hence, the reference gets modified. Notice that, we operate at individual pixel level to identify the valid interpolations. This becomes important in the context of effective reference selection. Consider the case of IG which selects a zero image as the reference. For some pixels in the input, a zero pixel may be in the gradient ascending direction. For others, it may not be so. For those pixels, our method would automatically discard the zero reference and its effective path will start from a valid (interpolated) point. Our Algorithm 1 is able to elegantly incorporate this without an explicit procedure to modify the reference set used by the original methods. We will clarify this in the camera ready version.
>
> > What exactly is the relevance of Lemma 1? What is different from standard Integrated Gradients, which also estimates the integral in eq. 3?
>
> Regarding Eq. 3, both our path and references are different from IG. We allow a whole set of references (i.e. $x’$ comes from $D_i$) as opposed to a single reference of IG. Also, our eventual path consists of valid interpolation points that are necessarily along the gradient ascending direction w.r.t. input – not just any point like IG. It is easy to imagine that integrating gradients using a potentially large reference set can be computationally intractable. Therefore, we introduce a method to approximate the integration in Section 4.2. Lemma 1 ensures that the procedure we use for the integration approximation is valid. It is a stepping stone towards computing the attributions.

---

> > ### Author Response · Authors · 2022-11-14
> > **Response to brex (2/2)**
> >
> > > The mathematical notation/naming is not optimal, which makes the method section difficult to read. Thanks for the comment, we will clarify the notation/naming. Our specific response is as follows.
> >
> > - In Eq. 3 (and other places), we will change the index variable of $D$.
> > - $x’$ is conventionally used in literature. Hence, we prefer to retain it. We will change the index variable of $D$.
> > - ‘Difference’ actually becomes even more confusing as a standalone word. Hence, we had eventually replaced it with ‘variation’. To clarify, we will explicitly mention that we mean difference.
> > - We must refer to gradient signs multiple times in our discussion. We used the notations in the set to keep the discussion concise.
> >
> > > What are the main differences to and advantages over Guided Integrated Gradients (GIG)?
> >
> > Our method is very different from GIG [1]. GIG is a heuristic-based method. Quoted from [1] “[GIG] selects the features [on the path] with the lowest absolute values of partial derivatives (e.g., bottom 10%), and moves only that subset closer to the intensity in the input image, leaving all others unchanged.” Our method does not need any thresholding due to its principled nature. We do not define subsets for a feature (to adapt them as GIG). Our idea of valid path features is entirely different. GIG also permits only a single reference and is not concerned about re-calibrating other path based methods. We permit multiple baselines, and contribute towards enhancing the broader path attribution strategy. We will explain this in camera ready.
> > > Additional minor issues.
> > - We will add AUC for the figures for a clear comparison. AUC are already given for Fig. 4 in the legend (values in the brackets).
> > - Since we have provided the comparison between IG-SQ and IG-SQ* in Figure 3 and Figure 5, we don’t show the baseline methods (e.g., IG-SQ) again but compare the method with other state-of-the-art attribution methods in Figure 4.
> > - In Fig 6, shaded area indicates the standard deviation employed in the Sensitivity metric. It will be mentioned.
> >
> > [1] Kapishnikov, Andrei, et al. "Guided integrated gradients: An adaptive path method for removing noise." Proceedings of the IEEE/CVF Conference on Computer Vision and Pattern Recognition. 2021.

---

> > ### Comment · Reviewer_brex · 2022-11-14
> > **Follow-up questions**
> >
> > I thank the authors for their detailed response, I will take it into account for the final recommendation.
> >
> > In order to give the authors sufficient time to answer, I would like to ask two follow-up questions right away, and might ask for additional clarification later on.
> >
> > **First**, I fully agree with the authors that Algorithm 1 should not be considered to be their only contribution and I understand that, being pseudo-code, some minor details might of course be neglected.
> > That said, I would nonetheless expect Algorithm 1 to accurately reflect the proposed method, to be consistent with the textual method description, and to help the reader to understand how the method works. However, I am currently not fully convinced that this is the case and would highly appreciate if the authors could clarify.
> >
> > In particular, I have the following questions and remarks.
> > 1) The authors say that only valid interpolation points are used. However, in Algorithm 1 the average gradient is computed over *all* interpolation points (lines 3 to 6) and the test for 'validity' is only done afterwards (line 7) for the *combined* result; as such, 'invalid' points can in fact influence the resulting attribution. This seems to be contradicting the authors' claim that their method "integrate[s] the gradients using only the valid interpolation points for each pixel". Which one of the two descriptions is correct?
> >
> > 2) The authors claim they do not employ ReLU. However, I would like to point out that lines 7 to 9 are equivalent to $M_i \leftarrow M_i + \text{ReLU}((x_i-x'_i)\times \bar{g})$, since $M_i$ is then only updated whenever $(x_i-x_i')\times\bar g_i$ is greater than 0. As such, the update in line 9 is in fact adding $\text{ReLU}(\text{IntGrad}\ (x; x', \theta))$ to $M$, where $\text{IntGrad}(x; x', \theta)$ computes the conventional integrated gradients with baseline $x'$ for a model with parameters $\theta$ and ReLU is done element-wise (i.e., per pixel).
> >
> > 3) Given the above remarks, in its current form the pseudo-code suggests that the re-calibrated attributions are computed as $\langle \text{ReLU}(\text{IntGrad}\ (x; x', \theta))\rangle_{\{x'\}}$ where $\langle \cdot \rangle_{\{x'\}}$ denotes the average over all $x'\in D$. This, in turn, is only a minor modification of the given baseline such as, e.g., IG-SG, with the only difference being the ReLU function (as the authors clarified in their answer, the reference set $D$ is the same as in any given baseline method). Is this indeed the case?
> >
> > 4) Finally, what does $\text{avg}(M_i)$ in line 10 do? As far as I can tell from the pseudo-code, $M_i$ is a single number and it is unclear to me what the average is computed over in this case. Probably related to this question: at what point does the importance sampling enter the picture?
> >
> >
> > **Secondly**, the notation of eq. (3) is still unclear to me. While the authors say that their method "allow[s for] a whole set of references", the reference set $D$ which appears on the left-hand side of the equation is not actually used on the right-hand side (RHS). Instead, a single $x'$ is used on the RHS, which is said to be equal to some $D_k$ (in the paper $D_i$) from the reference set. How exactly does the proposed method use the reference set? Judging from the pseudo-code, an averaging operation (over $D$) is missing from eq. 3.
> >
> > I am looking forward to the authors' response.

---

> > > ### Author Response · Authors · 2022-11-17
> > > **Response to Follow-up question (1/2)**
> > >
> > > Thank you for the follow up questions. We provide our response below.
> > >
> > >
> > > Response to the __first question__
> > >
> > > > I would nonetheless expect Algorithm 1 to accurately reflect the proposed method, to be consistent with the textual method description, and to help the reader to understand how the method works.
> > >
> > > Thank you for the suggestion. We will revise Alg. 1 to reflect the attribution estimation with valid interpolations. Notice that, in the original version, the algorithm was meant to present the procedure with valid reference. We will move that to the supplementary material and make a note about it. Algorithm 1 will be replaced by the following to cover the aspect of valid interpolations.
> > >
> > > ---
> > > Algorithm 1: Attribution Re-Calibration with Valid Interpolations
> > >
> > > ---
> > > **input:** Input $x$, input dimension $i$, reference set $D$, steps $n$. \
> > > **output:** Attribution $M_i$ \
> > > **initialize:** Attribution set $A_i=\emptyset$ \
> > > **for** each $x'$ in $D$ **do** $\color{gray}{// Iterate \space references.}$ \
> > > $\qquad$ **for** $k \leftarrow 1$ **to** $n$ **do** $\color{gray}{// Integral \space path.}$ \
> > > $\qquad\qquad \tilde{x} \leftarrow x'+\frac{k}{n}(x-x')$ \
> > > $\qquad\qquad \phi \leftarrow (x_i-\tilde{x}_i) \times \frac{\partial S_c(\tilde{x})}{\partial \tilde{x}_i}$ \
> > > $\qquad\qquad$ **if** $\phi \geq 0$ **then** \
> > > $\qquad\qquad\qquad$ $\color{gray}{// Itegrate \space attributions \space with \space valid \space interpolations.}$ \
> > > $\qquad\qquad\qquad A_i \leftarrow A_i \cup \\{\phi\\}$ \
> > > $\qquad\qquad$ **end if** \
> > > $\qquad$ **end for** \
> > > **end for** \
> > > $M_i \leftarrow \lambda \times$ avg($A_i$)
> > >
> > > ---
> > >
> > > > This seems to be contradicting the authors' claim that their method "integrate[s] the gradients using only the valid interpolation points for each pixel". Which one of the two descriptions is correct?
> > >
> > > Actually, the two descriptions are for valid interpolations and references respectively. Alg. 1 (in the original version) shows the pseudo-code for attribution estimation with valid references. In Section 4.3, we show that the attribution estimated with valid interpolations can be approximated with valid references. Therefore, the two descriptions are not contradictory.
> > >
> > > In section A.3.1 of the supplementary material, we conduct more experiments to compare the difference between valid references and interpolations. The experimental results in Fig.7&8 demonstrate that attributions calculated with valid references can efficiently achieve high performance. On the other hand, as the number of references grows, attribution calculated by valid interpolations can achieve higher performance since valid references may bring noise into the attribution estimation. The result demonstrates that the two strategies are consistent with our insights.
> > >
> > > > I would like to point out that lines 7 to 9 are equivalent to … and ReLU is done element-wise.
> > > > This, in turn, is only a minor modification … with the only difference being the ReLU function …
> > >
> > > Thank you for explaining further. However, there is a subtle point that makes our method different from ReLU. Unlike taking $\text{ReLU}$ to set all negative attributions to zero and then compute their average, Algorithm 1 only accumulates positive attributions and computes the average over the positive attributions. The two are actually different. The difference also impacts whether the attribution estimation can follow Lemma 1. In other words, only by averaging all attributions with identified references can an accurate attribution be estimated. We have also tested attribution with the ReLU activation. The results show that attributions with $\text{ReLU}$ activation actually cause a large performance degradation in comparison to our method. The performance gap demonstrates that the performance gain of our method does not come from ReLU-like activations. If we employ ReLU to rewrite Alg. 1, the pseudo-code would be $M_i \leftarrow M_i + \text{ReLU}((x_i - x’_i) \times \bar{g}) / L$ where L denotes the number of valid references for $x_i$.

---

> > > > ### Author Response · Authors · 2022-11-17
> > > > **Response to Follow-up question (2/2)**
> > > >
> > > > > What does avg … do?
> > > >
> > > > Thank you for this query. In fact, we made a mistake here, and we are highly grateful to the reviewer for raising the point. In Alg. 1, Line 10 should move outside the loop. As such, Line 10 computes the average of all the attributions estimated with different valid references followed by Lemma 1. $M_i$ represents the average of attributions estimated with different valid references. Considering that the use of $M_i$ may cause ambiguity, we now update the algorithm by introducing $A$ to represent the set of attributions estimated with valid references. The updated Algorithm for attribution estimation with valid ‘references’ is as follows.
> > > >
> > > > ---
> > > > Algorithm 2: Attribution Re-Calibration with Valid References
> > > >
> > > > ---
> > > > **input:** Input $x$, input dimension $i$, reference set $D$, steps $n$. \
> > > > **output:** Attribution $M_i$ \
> > > > **initialize:** $A_i=\emptyset$ \
> > > > **for** each $x'$ in $D$ **do** $\color{gray}{// Iterate \space references.}$ \
> > > > $\qquad$ **for** $k \leftarrow 1$ **to** $n$ **do** $\color{gray}{// Integral \space path.}$ \
> > > > $\qquad\qquad \tilde{x} \leftarrow x'+\frac{k}{n}(x-x')$ \
> > > > $\qquad\qquad \color{gray}{// Average \space gradients.}$ \
> > > > $\qquad\qquad \bar{g} \leftarrow \bar{g} +  \nabla_{\tilde{x}}/n $ \
> > > > $\qquad$ **end for** \
> > > > $\qquad$ **if** $\bar{g}_i \cdot (x_i-x'_i) \geq 0$ **then** \
> > > > $\qquad\qquad \color{gray}{// Itegrate \space attributions \space with \space valid \space references.}$ \
> > > > $\qquad\qquad A_i \leftarrow A_i \cup \\{\bar{g} \times (x_i-x'_i)\\}$ \
> > > > $\qquad$ **end if** \
> > > > **end for** \
> > > > $M_i \leftarrow \lambda \times$ avg($A_i$)
> > > >
> > > > ---
> > > >
> > > >
> > > >
> > > > > … what point does the importance sampling enter the picture?
> > > >
> > > > Importance sampling is employed to convert the attribution estimation with valid interpolations to valid references. In Section 4.3 and lemma 2, we show the feasibility of this transformation with importance sampling.
> > > >
> > > >
> > > > Response to the __second question__
> > > > > the notation of eq. (3) is still unclear to me. … How exactly does the proposed method use the reference set?
> > > >
> > > > The main idea is that the attribution of different pixels should be estimated with different references and integral paths. For an input pixel $x_i$ (along the $i$-th dimension), Eq. 3 defines the attribution estimation for $x_i$ with the reference $x’=D_i$ and integral path $\gamma_i$. In Eq. 3, the dimension index $i$ is corresponding to both the reference index and integral path index. Since it is intractable to find the reference set, the rest sections employ $D$ to represent an arbitrary reference set. In Alg.1 the reference set $D$ also represents a random reference set from existing methods. We will clarify this point in the revision.
> > > >
> > > > Regarding the difference between Eq. 3 and Alg. 1, it should be noted that we employ the differential of the path $d\alpha$ instead of the input $dx$. This means that Eq 3 can be reformulated as follows:
> > > > $
> > > > \int_{\alpha=0}^1 \frac{\partial S_c(\tilde{x})}{\partial \tilde{x}_i} \mathrm{d}\alpha \
> > > > = \sum_1^n \frac{\partial S_c(\tilde{x})}{\partial \tilde{x}_i} / n
> > > > $
> > > >
> > > > Therefore, the formula is actually according to the pseudo-code.

---

### Decision · Program_Chairs · 2023-01-20

**Decision:**

Accept: notable-top-25%

**Justification For Why Not Higher Score:**

It is not well written.

**Justification For Why Not Lower Score:**

It has meaningful contributions

**Metareview: Summary, Strengths And Weaknesses:**

This article addresses the problem on explainability. The proposed method is well motivated while some of the assumptions do not hold in practice. So, the paper proposes some extensions to overcome it. The method consistently improves on several benchmarks. The reviewers agree that the evaluation is correct and extensive. They address important shortcomings of existing methods. The paper is clear even though it can be improved.

The reviewers pointed out several issues like lack of clarity which is one of the main issues. Also, the paper does not comment on limitations.

The authors and reviewers had a long discussion and improved the paper. Still, it requires some iterations to be better written. I highly recommend the authors to improve it.

**Note From Pc:**

if the above contains the word "oral" or "spotlight" please see: "oral" presentation means -> notable-top-5% and "spotlight" means -> notable-top-25%. As stated in our emails, we are disassociating presentation type from AC recommendations